# Mechanical self-adaptive porous valve relying on surface tension for energy harvesting from low-flux bubbles

Yu Du [1] ✉, Ping Li [2,3], Yumei Wen [2,3], Yunsheng Fan [1] & Zhichen Liu [1]

Harvesting energy from subsea bubbles, such as those produced by photosynthesis of benthic plants or submarine methane seepage, is a promising solution for powering subsea environment perception devices, but the low gas flux brings significant challenges. Herein, we propose a passive mechanical self-adaptive porous valve with self-adaptive mechanical properties and high gas permeability. It improves energy harvesting performance from low-flux bubbles by controlling bubble accumulation and high-speed release. Unlike traditional active mechanical metamaterials, this passive design utilizes gas-liquid interface deformation (rather than metamaterial actuation) to generate self-adaptive Laplace pressure counteracting bubble buoyancy, and thus requires no external energy. The porous valve has a stable opening threshold inversely proportional to its structural pore diameter. Compared with a bubble energy harvesting device with no valve, the instantaneous gas-intake rate of the device equipped with the porous valve is increased by one to four orders of magnitude, and the maximum output power and electrical energy production are enhanced by factors of 36.6 and 16.4, respectively. The energy of underwater biological metabolic gas with a low flux ($28\,\mu L \cdot min^{-1}$) is effectively harvested and supplied to an underwater sensor. This work is expected to provide in situ energy for subsea self-powered sensing and autonomous exploration.

The ocean covers 71% of the Earth's surface and is rich in natural resources and energy[1–3]. Scientific tasks such as the exploration of deep-sea resources and the maintenance of the marine environment and ecological security have promoted the development of in-situ seabed observation technologies[4,5]. Numerous distributed in situ sensing devices required for subsea observation are difficult to be connected to the power grid. Directly obtaining energy from the subsea environment is a crucial way to solve the problem of in situ self-powering[6–8]. Conventional marine energy sources, such as ocean wave energy[9,10], tidal energy[11], ocean current energy[12], ocean thermal energy[13], and salinity-gradient energy[14,15], are usually distributed in the

shallow layers of the ocean or specific sea areas, and are difficult to supply the self-powered sensing devices on the deep-sea floor. Unlike the above-mentioned energy sources, subsea bubbles contain abundant potential energy and are distributed widely[16–19]. The photosynthesis of marine plants contributes 50% of the oxygen on the Earth[20]; the seabed releases up to 48 Tg of methane annually[21]. As numerous subsea bubbles rise, the work performed by gas buoyancy drives the surrounding liquid to form an ascending gas-liquid two-phase flow. This process releases a large amount of potential energy, which is then converted into fluid kinetic energy[22–24]. Calculated based on the average depth of 3700 m of the global ocean, if 48 Tg of

[1]College of Marine Electrical Engineering, Dalian Maritime University, Dalian, Liaoning, China. [2]School of Automation and Intelligent Sensing, Shanghai Jiao Tong University, Shanghai, China. [3]State Key Laboratory of Submarine Geoscience, Shanghai Jiao Tong University, Shanghai, China. ✉e-mail: yudu@dlmu.edu.cn

methane bubbles rise from the seabed to the water surface, the released buoyancy potential energy will be as high as 10.8 TWh. Efficient collection of bubble potential energy is expected to solve the problem of subsea in situ self-powering of numerous underwater sensing devices.

In actual marine environments, bubbles are usually released dispersedly from multiple points, which leads to a low gas flux of the seabed per unit area[25,26]. Specifically, the gas production volume per minute through the photosynthesis of marine plants per unit area is usually on the order of microliters[27], and the gas flux at a single subsea seepage point can be as low as a few milliliters per minute. For example, the rate of oxygen-rich bubbles produced by microalgae in the waters of St. Joseph Bay in the Gulf of Mexico is only $225 \pm 132$ mL per square meter per day[28]. The gas production rates through the photosynthesis of a single leaf of two seagrasses, H. wrightii and S. filiforme, are only $24.2 \, \mu L \cdot min^{-1}$ and $36.0 \, \mu L \cdot min^{-1}$, respectively[29]. The gas fluxes at the two representative observed seepage points in the Site F cold-spring activity area in the South China Sea are $6.75 \, mL \cdot min^{-1}$ and $2.18 \, mL \cdot min^{-1}$, respectively[30]. However, the power generation devices in any bubble energy harvesting system all have working thresholds. It is essential to rely on a sufficiently high gas-intake rate to generate a high-speed fluid, for the purpose of providing a force large enough to drive mechanisms such as turbines. Some devices with low starting thresholds require a gas-intake rate of more than $40 \, mL \cdot min^{-1}$[31], and some with higher thresholds even require a gas-intake rate as high as $1 \, L \cdot min^{-1}$[32]. Therefore, the low-flux bubbles widely existing in the actual subsea environment can hardly meet the working requirements of the energy harvesting devices.

To date, there are mainly two potential approaches for harvesting energy from low-flux bubbles. One is to reduce the working threshold of the energy harvester itself, and the other is to enhance the instantaneous gas-intake rate of the energy harvesting device. For the first approach, the utilization of non-mechanical-type energy harvesters instead of traditional rotary mechanical energy harvesters can significantly lower the working threshold[33]. Such energy harvesters usually operate based on the principle of electrostatic induction, using triboelectric materials or electret materials with surface charges as transduction elements. When the gas-liquid interface moves on the surface of the charged materials, the variation of the electrostatic field distribution can drive the reciprocating flow of electrons in the external circuit, thereby outputting an alternating current signal. Utilizing this basic principle, multiple energy converters have been proposed for low-flux bubble energy harvesting, such as dual-electrode and multi-electrode bubble energy harvesters with hydrophobic surfaces containing embedded charges (hydrophobic electrets)[34,35], electret bubble energy generators inspired by transistor[36,37], and triboelectric nanogenerators based on solid-liquid friction[38–42]. Although these energy harvesters, based on electrostatic induction, have a low working threshold, they universally suffer from surface charge decay. Notably, this decay process is accelerated specifically when the devices operate in conductive salt solutions. The output voltage of the electrostatic energy harvesters drops rapidly with the increase of salt concentration[40], and they even fail to produce an effective output under the salt concentration of ordinary seawater. Due to this issue, they are currently more suitable for working in freshwater environments, and are difficult to be used as energy harvesters in a subsea bubble energy harvesting system.

For the second approach, to improve the instantaneous gas-intake rate of the energy harvesting device, it is a feasible scheme to accumulate enough bubbles before releasing them[32], which requires effective on-off control of the fluid flow channels. In the field of fluid control, mechanical metamaterials rely on the specially designed internal geometric structures to acquire functions that natural materials cannot possess, and are widely used as actuating devices[43]. Driven by external light sources[44], magnetic fields[45], heat sources[46], or

mechanical forces[47], the structures of mechanical metamaterials deform, thus squeezing the flow channels and achieving fluid control. For example, a re-entrant mechanical metamaterial with a "bowtie" structure is wrapped around the outside of flexible fluid channels. When pressure is applied, the metamaterial allows for the controlled propagation of mechanical forces through the structure by redirecting the pressure around the flow channels, thereby controlling the "always open" or "closeable" state of the fluid channels[43]. It should be noted that traditional mechanical metamaterials work relying on the deformation of their own structures. Therefore, they inevitably require external forces to do work and consume energy. However, as active actuators, the complex external control and substantial energy consumption make it difficult for traditional mechanical metamaterials to be applied in the bubble energy harvesting systems for low-flux gas sources in subsea environments. Therefore, to achieve bubble accumulation and high-speed release under passive conditions, it is necessary to develop a passive mechanical structure that does not rely on its own structural deformation.

Herein, we propose a passive porous interface mechanical structure with high gas permeability and self-adaptive interface mechanical properties. On this basis, a corresponding mechanical self-adaptive porous valve is fabricated to enhance the energy harvesting performance of low-flux bubbles. Completely different from the fluid-control mechanism of traditional active mechanical metamaterials, the proposed passive porous interface mechanical structure relies on a specially designed array of conical capillary micropores to induce the deformation of the gas-liquid interfaces within the micropores, without the need for the deformation of the material structure itself, and thus consumes no energy. Under the effect of liquid surface tension, the adaptive deformation of the convex gas-liquid interface can provide Laplace pressure to resist the buoyancy of bubbles, achieving the passive and automatic accumulation of low-flux bubbles. Parallel bubble release via the micropore array, combined with positive-feedback interfacial force, is used to significantly enhance the instantaneous gas-intake rate and the rotational speed-flux coefficient of the energy harvesting device. First, we study the threshold characteristics of the porous valve through mechanical analysis and experiments on the gas-liquid interface of the porous interface mechanical structure. Then, bubble energy harvesting experiments based on the porous valve are carried out to demonstrate the contributions of the porous valve in improving the instantaneous gas-intake rate, output power, output energy density, etc. of the bubble energy harvesting device (BEHD). Finally, an experimental demonstration of energy harvesting and utilization of low-flux bubbles generated by photosynthesis of aquatic plants is conducted to verify the significant advantages of the porous valve in low-flux bubble energy harvesting. The demonstrations of the mechanical self-adaptive porous valve based on the passive porous interface mechanical structure confirm its promising application prospects in bubble flow rate regulation, energy harvesting, and the utilization of subsea low-flux bubbles.

## Results
### Design and operational mechanism
In the subsea environment, whether it is the photosynthetic bubbles produced by marine plants or the bubbles released by submarine methane seepage, they are all slowly released from multiple scattered points, and thus, the gas flux of bubbles released per unit area of the seabed is low. Figure 1a presents the working scenario of the mechanical self-adaptive porous valve made of the passive porous interface mechanical structure. The low-flux bubbles are first collected and accumulated beneath the porous valve before entering the BEHD. The BEHD consists of a bubble rising pipe and a turbine generator. When the accumulated gas attains the threshold of the porous valve, the porous valve immediately opens and releases the accumulated gas at a high speed within a short period of time. The released bubbles flow

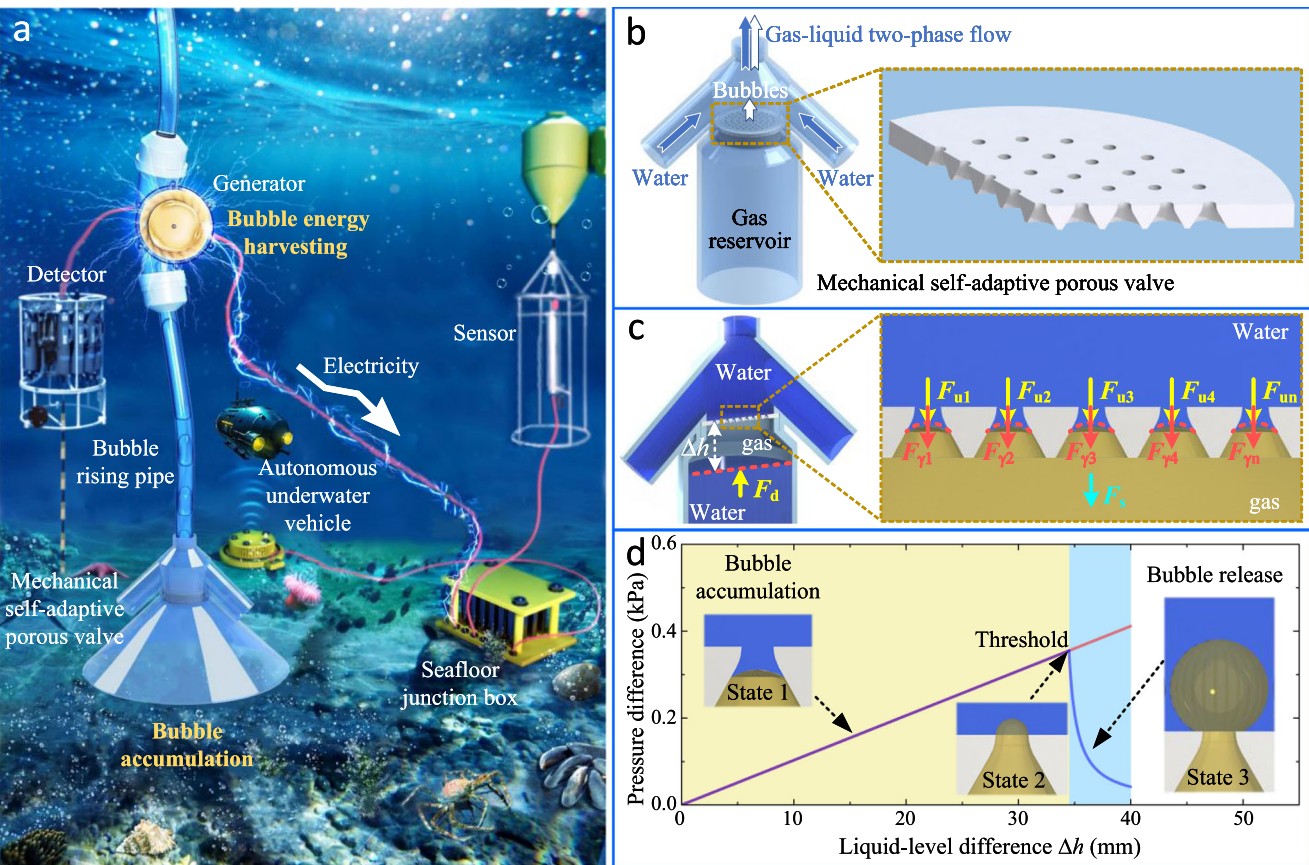

**Fig. 1 | Application scenarios, structure, and mechanical principles of the mechanical self-adaptive porous valve made of the passive porous interface mechanical structure. a** Conceptual diagram of the porous valve used for low-flux bubble accumulation and high-speed release in subsea bubble energy harvesting. **b** Overall structure of the porous valve and sectional structure of the passive porous interface mechanical structure. **c** Analysis of the interfacial forces inside the porous valve. The red dashed line represents the gas-liquid interface, the yellow arrows represent the liquid pressure, the red arrows represent the Laplace pressure, the cyan arrow represents the downward pressure applied to the gas by the gas-solid interface, and $\Delta h$ is the liquid-level difference between the gas-liquid interface above the gas and that below the gas. **d** Self-adaptive dynamic adjustment of the interfacial forces under the action of liquid surface tension. The red curve represents the variation trend of the liquid pressure difference acting on the gas below the porous plate with the liquid-level difference $\Delta h$, while the blue curve represents the variation trend of the Laplace pressure difference $\Delta p$ at the convex gas-liquid interface in each micropore with the liquid-level difference $\Delta h$.

into the bubble rising pipe positioned above the porous valve. Buoyancy from the bubbles propels the water in the pipe upward, creating a rising gas-liquid two-phase flow that in turn drives the turbine generator to rotate and output electrical energy. The seawater from the external environment flows into the pipe through the water inlets on both sides of the porous valve and finally flows out from the pipe's upper end above the generator, thereby establishing water circulation. During this process of water flow circulation, as discussed in ref. 48, the buoyancy potential energy of the bubbles undergoes two successive conversions: first into the kinetic energy of the gas-liquid two-phase fluid, and then into electrical energy via the generator. The electrical energy output by the device can be stored in the subsea energy storage device and supplied to various sensors and exploration devices in the subsea observation network via seafloor junction boxes, or used to wirelessly charge autonomous underwater vehicles (AUV).

The working principle of the porous valve is as follows: it relies on the surface tension of the convex gas-liquid interfaces inside the aerophobic passive porous structure to generate a self-adaptively changing Laplace pressure, and utilizes the interaction between Laplace pressure and liquid pressure to achieve bubble accumulation and high-speed release. The structure of the porous valve is shown in Fig. 1b. Its core component is the proposed passive porous interface mechanical structure, which is a circular thin plate full of perforative conical micropores and embedded within the porous valve. All the micropores have the same shape and size, being conical with a small

radius $r_u$ at the upper port and a large radius $r_d$ at the lower port. The gas reservoir below the porous plate has an open lower rim, permitting buoyancy-driven entry of bubbles from below. There are water inlets on both sides of the gas-liquid mixing chamber above the porous plate. The water from the external environment can flow in from the water inlets on both sides and enter the bubble rising pipe along with the bubbles released by the porous valve. The surface of the porous plate is treated to be hydrophilic to exhibit aerophobic characteristics in water.

The interfacial mechanical principles of bubble accumulation and high-speed release in the porous valve are shown in Fig. 1c, d. According to the analysis method for interfacial forces in ref. 31, the gravity of the accumulated gas is negligible, and all the vertical external forces on the gas come from the surrounding gas-liquid and gas-solid interfaces. As shown in Fig. 1c, due to the aerophobic characteristics of the hydrophilic surface of the porous plate, an upward convex gas-liquid interface is generated in each conical micropore above the gas. The effect of surface tension produces a downward Laplace pressure $F_{\gamma i}$ ($i = 1, 2, \cdots, n$) on each convex gas-liquid interface, where $n$ represents the number of the micropores. Liquid pressure exerts a downward force $F_{ui}$ ($i = 1, 2, \cdots, n$) on each convex gas-liquid interface above the gas, while applying an upward liquid pressure $F_d$ to the horizontal gas-liquid interface below the gas. In addition, there is also a resultant downward pressure $F_s$ applied to the gas by the gas-solid interface made up of the lower surface of the porous plate and the inner wall of

the gas reservoir. The upward resultant force acting on the gas in the porous valve is

$$\sum F = F_{\mathrm{d}} - \sum_{i=1}^{n} F_{\mathrm{u}i} - \sum_{i=1}^{n} F_{\gamma i} - F_{\mathrm{s}}. \tag{1}$$

The liquid pressure at the convex gas-liquid interface in micropore $i$ above the gas and that at the horizontal gas-liquid interface below the gas are respectively

$$F_{\mathrm{u}i} = p_{\mathrm{u}i} S_i, \tag{2}$$

$$F_{\mathrm{d}} = p_{\mathrm{d}} S_{\mathrm{d}}, \tag{3}$$

where $p_{\mathrm{u}i}$ represents the liquid pressure intensity acting on the convex gas-liquid interface in micropore $i$; $p_{\mathrm{d}}$ represents the liquid pressure intensity acting on the horizontal gas-liquid interface below the gas; $S_i$ and $S_{\mathrm{d}}$ represent the effective area (projected area in the vertical direction) of the convex gas-liquid interface in micropore $i$ and the horizontal gas-liquid interface below the gas, respectively. The Laplace pressure acting on the convex gas-liquid interface in micropore $i$ above the gas is

$$F_{\gamma i} = \Delta p_i S_i, \tag{4}$$

where $\Delta p_i$ represents the Laplace pressure difference generated by the surface tension at the convex gas-liquid interface in micropore $i$. The resultant pressure $F_{\mathrm{s}}$ exerted on the gas by the gas-solid interface is the reaction to gas pressure; and thus the pressure intensity $p_{\mathrm{s}}$ that the gas-solid interface exerts on the gas matches the gas pressure intensity $p_{\mathrm{g}}$. Moreover, the gas pressure intensity $p_{\mathrm{g}}$ also equals the liquid pressure intensity $p_{\mathrm{d}}$, as no Laplace pressure difference exists across the horizontal gas-liquid interface below the gas. Therefore, $p_{\mathrm{s}} = p_{\mathrm{g}} = p_{\mathrm{d}}$, and the resultant pressure exerted on the gas by the gas-solid interface is

$$F_{\mathrm{s}} = p_{\mathrm{s}} \left( S_{\mathrm{d}} - \sum_{i=1}^{n} S_i \right) = p_{\mathrm{d}} \left( S_{\mathrm{d}} - \sum_{i=1}^{n} S_i \right). \tag{5}$$

Substitution of Eqs. (2), (3), (4), and (5) into Eq. (1) allows it to be rewritten as

$$\sum F = p_{\mathrm{d}} \sum_{i=1}^{n} S_i - \sum_{i=1}^{n} p_{\mathrm{u}i} S_i - \sum_{i=1}^{n} \Delta p_i S_i. \tag{6}$$

When the parameters of the micropores on the porous plate are consistent, the convex gas-liquid interface within each micropore has the same shape and parameters. Accordingly, each convex gas-liquid interface is subject to the same liquid pressure intensity $p_{\mathrm{u}}$ and the same Laplace pressure difference $\Delta p$, that is, $p_{\mathrm{u}i} = p_{\mathrm{u}}$ and $\Delta p_i = \Delta p$ ($i = 1, 2, \cdots, n$). Then Eq. (6) can be simplified to

$$\sum F = (p_{\mathrm{d}} - p_{\mathrm{u}} - \Delta p) \sum_{i=1}^{n} S_i. \tag{7}$$

During bubble accumulation, the gas residing inside the porous valve is maintained in a dynamic equilibrium state, and the resultant force in the vertical direction is $\sum F = 0$. By substituting this condition into Eq. (7), it can obtain that

$$p_{\mathrm{d}} - p_{\mathrm{u}} = \Delta p. \tag{8}$$

Equation (8) indicates that during the bubble-accumulation stage, the liquid pressure difference $p_{\mathrm{d}} - p_{\mathrm{u}}$ exerted on the accumulated gas

always equals the Laplace pressure difference $\Delta p$ on the convex gas-liquid interface within each micropore, regardless of the number of micropores. The hydrostatic pressure equation and Young–Laplace equation are respectively

$$p_{\mathrm{d}} - p_{\mathrm{u}} = \rho g \Delta h, \tag{9}$$

$$\Delta p = \frac{2\gamma}{R}, \tag{10}$$

where $\rho$ (liquid density), $g$ (gravitational acceleration), and $\gamma$ (surface tension of the liquid) are physical parameters determined by the liquid's intrinsic properties and its surrounding environment. $\Delta h$ (liquid-level difference between the gas-liquid interface above the gas and that below the gas) and $R$ (radius of curvature of the convex gas-liquid interface) are geometric parameters that vary with the bubble accumulation and release process. According to Eqs. (8)–(10) and the corresponding parameters ($\gamma = 7.1 \times 10^{-2}\,\mathrm{N\,m^{-1}}$, $\rho = 1.05 \times 10^{3}\,\mathrm{kg\,m^{-3}}$, $g = 9.8\,\mathrm{m\,s^{-2}}$), the variation curves of $p_{\mathrm{d}} - p_{\mathrm{u}}$ and $\Delta p$ corresponding to the bubble-accumulation stage (State 1) are plotted in Fig. 1d with light yellow background. As shown in Fig. 1d, the radius of curvature $R$ undergoes a gradual decrease as bubbles accumulate, which in turn causes the Laplace pressure difference $\Delta p$ to rise gradually. $\Delta p$ and the liquid pressure difference $p_{\mathrm{d}} - p_{\mathrm{u}}$ increase synchronously with $\Delta h$ and maintain a dynamic equilibrium, preventing the release of bubbles. When the three-phase contact line within the micropore reaches the upper port of the micropore (State 2), $R$ decreases to the minimum value $R_{\min}$, which is equal to the radius $r_{\mathrm{u}}$ of the upper port of the micropore. At this time, $\Delta p$ reaches the maximum value $\Delta p_{\max} = 2\gamma/r_{\mathrm{u}}$ and can no longer counteract the increase of $p_{\mathrm{d}} - p_{\mathrm{u}}$. The porous valve reaches the threshold for bubble release, and the liquid-level difference in the porous valve is

$$\Delta h_{\mathrm{T}} = \frac{p_{\mathrm{d}} - p_{\mathrm{u}}}{\rho g} = \frac{\Delta p_{\max}}{\rho g} = \frac{2\gamma}{\rho g r_{\mathrm{u}}}. \tag{11}$$

$\Delta h_{\mathrm{T}}$ is defined as the opening threshold of the porous valve. Equation (11) shows that the opening threshold $\Delta h_{\mathrm{T}}$ of the porous valve is inversely proportional to the radius $r_{\mathrm{u}}$ (or diameter $d_{\mathrm{u}}$) of the upper port of the micropore.

Once the liquid-level difference $\Delta h$ of the accumulated gas reaches the opening threshold $\Delta h_{\mathrm{T}}$, continued entry of bubbles into the gas reservoir will cause $\Delta h$ to keep increasing, eventually exceeding the threshold and triggering a continuous rise in the liquid pressure difference. Meanwhile, as shown in the light blue background area in Fig. 1d and the illustration of state 3, the bubbles on the upper surface of the porous plate expand. The expansion of each bubble leads to an increase in the radius of curvature $R$, resulting in a significant decrease in the Laplace pressure difference $\Delta p = 2\gamma/R$. Therefore, a resultant vertical upward external force is instantaneously generated and exerted on the gas:

$$\sum F = \left( \rho g \Delta h - \frac{2\gamma}{R} \right) \sum_{i=1}^{n} S_i > 0. \tag{12}$$

The rapid increase of the resultant vertical upward external force enables the porous valve to enter the open state. The gas accumulated inside the porous valve is simultaneously released through numerous micropores. It means that the porous valve enters the stage of high-speed bubble release. As discussed in reference[31], when the released bubbles flow into the bubble rising pipe of the energy harvesting device, the increase in the flow rate of the rising gas-liquid two-phase flow in the pipe leads to a decrease in the liquid pressure intensity $p_{\mathrm{u}}$ above the porous plate, which in turn causes the liquid pressure difference $p_{\mathrm{d}} - p_{\mathrm{u}}$ to further increase, thereby providing a positive

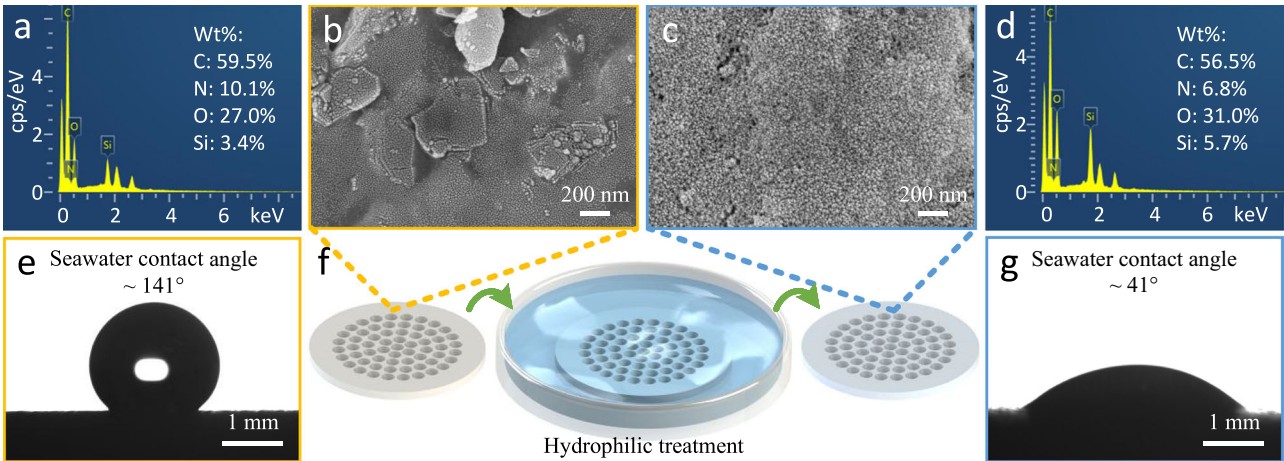

**Fig. 2 | Hydrophilic treatment, surface morphology and wettability of the porous interface mechanical structure in the mechanical self-adaptive porous valve. a** Surface elemental energy spectrum of the porous plate before the hydrophilic treatment (Wt% represents weight percentage). **b** Scanning electron microscope (SEM) image of the porous plate before the hydrophilic treatment. **c** SEM image of the porous plate after the hydrophilic treatment. **d** Surface elemental energy spectrum diagram of the porous plate after the hydrophilic treatment. **e** Contact angle of seawater on the surface of the porous plate before the hydrophilic treatment. **f** Schematic diagram of the hydrophilic treatment process. **g** Contact angle of seawater on the surface of the porous plate after the hydrophilic treatment.

mechanical feedback effect for the high-speed release of bubbles through numerous micropores.

The working principle of the mechanical self-adaptive porous valve is summarized as follows. The porous valve relies on the designed conical micropore array to induce the automatic deformation of the upwardly convex gas-liquid interface inside numerous micropores while moving upward. Relying on the surface tension and the self-adaptive deformation of the gas-liquid interface, a self-adaptively changing Laplace pressure in the downward direction is generated. As bubbles accumulate, the gas-liquid interface undergoes an adaptive decrease in its radius of curvature, with the Laplace pressure difference increasing gradually as a consequence. The Laplace pressure difference maintains a dynamic mechanical equilibrium with the liquid pressure difference, thereby preventing the release of the accumulated gas. During bubble release, the increased gas-liquid interface curvature radius reduces the Laplace pressure difference rapidly, preventing it from countering the continued increase of the liquid pressure difference. The positive-feedback interface mechanical effect enables the bubbles to be released at a high speed. The prerequisite for realizing this interface mechanical effect is that the surface of the porous plate exhibits aerophobic characteristics, thereby generating an upwardly convex gas-liquid interface and a downward Laplace pressure.

## Surface hydrophilic treatment

To enable the surface of the porous interface mechanical structure to exhibit an aerophobic state, its surface was treated with hydrophilic modification. Figure 2 presents the effect of the hydrophilic treatment. The surface morphology and elemental composition of the porous plate before and after the hydrophilic treatment were tested by using a field emission scanning electron microscope (Sigma 500, Carl Zeiss Microscopy GmbH) equipped with an Oxford energy dispersive spectrometer. The contact angles of 5-μL seawater droplets (with a salinity of 3.5%) on the surface of the porous plate before and after the hydrophilic treatment were measured by using a contact angle meter (SDC-80B, Dongguan SINDIN Precision Instrument Co., Ltd).

Figure 2f shows the hydrophilic treatment process. The porous plate manufactured by 3D printing was immersed in a glass culture dish filled with commercial liquid hydrophilic agent (the main hydrophilic component is nano-silica, and the solvent is absolute ethanol). The glass culture dish was left to stand at room temperature until all

the absolute ethanol had evaporated and the surface of the porous plate was completely air-dried. Figure 2a–d shows the surface morphology and energy spectrum diagrams of the porous plate before and after the hydrophilic treatment. After the hydrophilic treatment, a layer of nano-silica is deposited on the surface of the porous plate (Fig. 2c), resulting in a visible change in the surface elemental composition. As shown in Fig. 2a, d, among the four elements of carbon, nitrogen, oxygen, and silicon, the weight percentages of silicon and oxygen increase from 3.4% and 27.0% to 5.7% and 31.0%, respectively. The hydrophilic treatment reduces the contact angle of seawater on the surface of the porous plate from ~141° to ~41°, indicating that the surface of the porous plate exhibits favorable wettability towards seawater. As shown in Fig. S2 (Supplementary information), the contact angle of a bubble on the surface of the porous plate immersed in seawater is ~146°, demonstrating that the porous plate exhibits favorable aerophobic characteristics in seawater after the hydrophilic treatment.

## Threshold characteristic

The aerophobic surface of the porous plate provides a necessary condition for the porous valve to generate downward Laplace pressure. To further investigate the effects of bubble accumulation and high-speed release of the porous valve, as well as its threshold characteristic, a high-speed camera was used to record the bubble accumulation and high-speed release processes of the porous valve. In the experiment, a porous valve with a pore diameter of 0.8 mm was installed at the lower end of the bubble rising pipe in the BEHD shown in Fig. S4 (Supplementary information).

Figure 3 shows the accumulation and high-speed release process of the bubbles in the porous valve. The corresponding physical phenomena are shown in Supplementary Movie 1. During the first 13.7 s, as shown in Fig. 3a–c, the porous valve is in the bubble-accumulation stage. Bubbles continuously enter the gas reservoir below the porous plate, resulting in a gradual drop of the liquid level in the gas reservoir. With the accumulation of gas below the porous plate, as shown in Supplementary Movie 2, the gas-liquid interfaces in numerous micropores gradually bulge upward. During this process, the radius of curvature of the gas-liquid interface in each micropore gradually decreases, leading to the synchronous increase of the downward Laplace pressure difference along with the upward liquid pressure difference (Fig. 1d). The Laplace pressure difference counteracts the

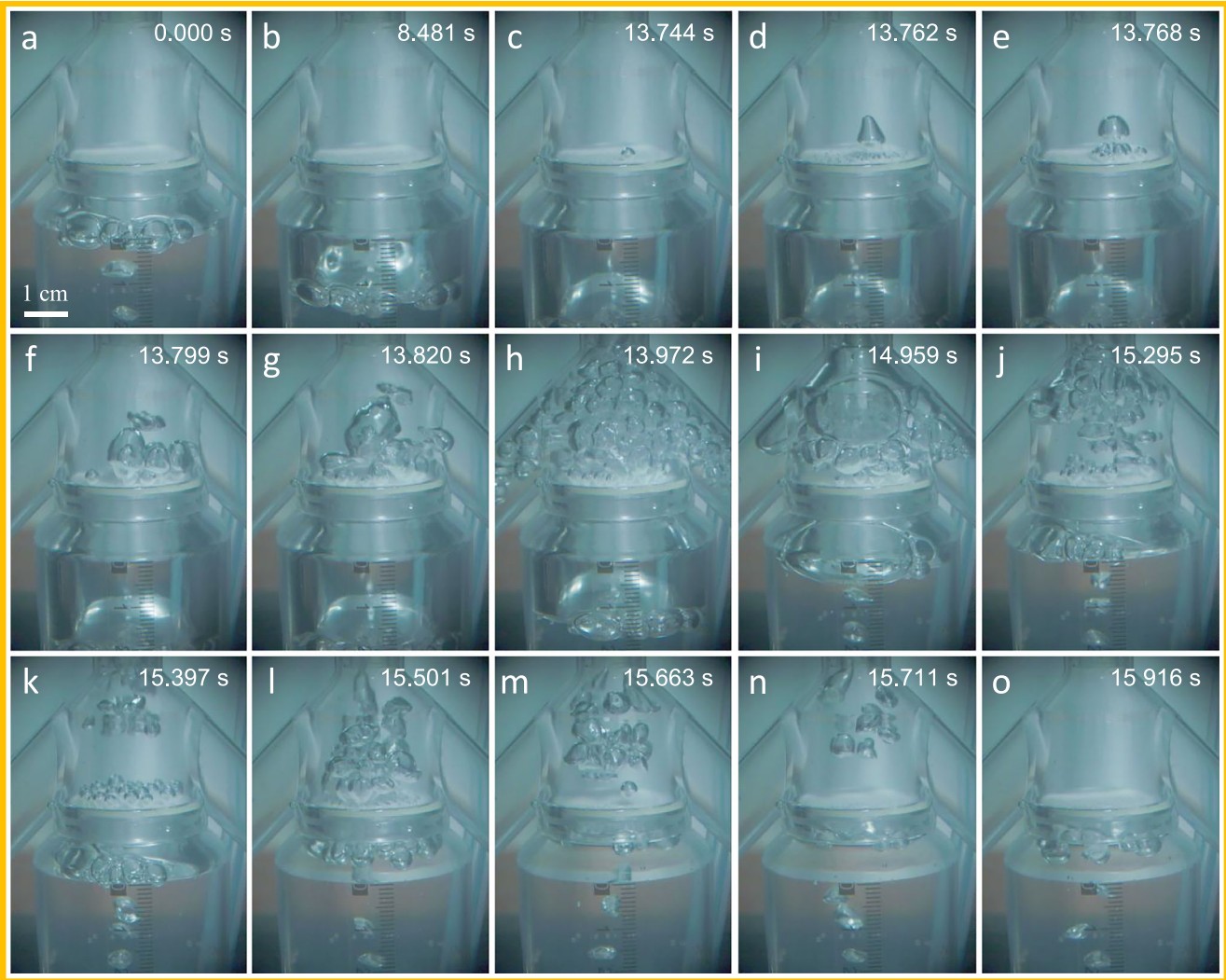

**Fig. 3 | Accumulation and high-speed release of the bubbles in the porous valve.** Photographs of the bubble in the porous valve at **a** 0.000 s, **b** 8.481 s, **c** 13.744 s, **d** 13.762 s, **e** 13.768 s, **f** 13.799 s, **g** 13.820 s, **h** 13.972 s, **i** 14.959 s, **j** 15.295 s, **k** 15.397 s, **l** 15.501 s, **m** 15.663 s, **n** 15.711 s, and **o** 15.916 s.

effect of the liquid pressure difference (bubble buoyancy), and thus the gas cannot be released through the porous plate. At 13.744 s, as shown in Fig. 3c, the liquid-level difference corresponding to the accumulated gas in the gas reservoir reaches the opening threshold of the porous valve. The accumulated gas then begins to break through the constraint of the Laplace pressure difference and is released through the porous plate into the gas-liquid mixing chamber. At this point, the porous valve enters the bubble-release stage.

Figures 3c–m show the process of the high-speed release of the gas accumulated inside the porous valve through numerous micropores. At 13.744 s (Fig. 3c), the first bubble is released from one micropore of the porous plate. At 13.762 s (Fig. 3d), this bubble detaches from the surface of the porous plate and becomes an independent bubble. Subsequently, many other micropores also start to release bubbles simultaneously, and thus the number of bubbles above the porous plate increases rapidly (Fig. 3e–h). At 13.972 s (Fig. 3h), numerous released bubbles gather above the porous plate and fill the entire gas-liquid mixing chamber. At 14.959 s (Fig. 3i), the gathered bubbles merge to form large bubbles and begin to enter the bubble rising pipe. As shown in Fig. 3h–m, the high-speed release of bubbles enables the gas in the gas reservoir to decrease rapidly. After mixing with water, the bubbles enter the bubble rising pipe above the porous valve, forming a rising gas-liquid two-phase flow. A gradual increase in the flow rate of the fluid above the porous plate reduces the liquid

pressure intensity $p_u$ above the porous plate, which in turn further increases the liquid pressure difference $p_d - p_u$ and accelerates the release of bubbles. By virtue of the positive feedback effect, as discussed in reference[31], after the liquid-level difference $p_d - p_u$ decreases to less than the opening threshold of the porous valve, the behavior of bubble release continues, thus ensuring the sufficient release of the accumulated gas. In addition, the decrease in the fluid pressure intensity above the porous plate causes the bubbles to be quickly sucked into the bubble rising pipe after being released and no longer gather in the gas-liquid mixing chamber (Fig. 3j–l). As shown in Fig. 3m, at 15.633 s, the last bubble leaves the surface of the porous plate, which means the end of the bubble-release stage. Almost all the gas accumulated in the gas reservoir is released, and the porous valve commences entering the bubble-accumulation stage of the subsequent cycle. At 15.916 s (Fig. 3o), all the bubbles released by the porous valve have entered the bubble rising pipe. There are no bubbles in the gas-liquid mixing chamber above the porous plate, which is filled with liquid.

The physical phenomena shown in Fig. 3 demonstrate the favorable performance of the mechanical self-adaptive porous valve based on the proposed passive porous interface mechanical structure. After a long period of accumulation lasting over 13.7 s, the accumulated gas was almost completely released within just 1.9 s, which proves that the porous valve exhibits a favorable capability of bubble accumulation

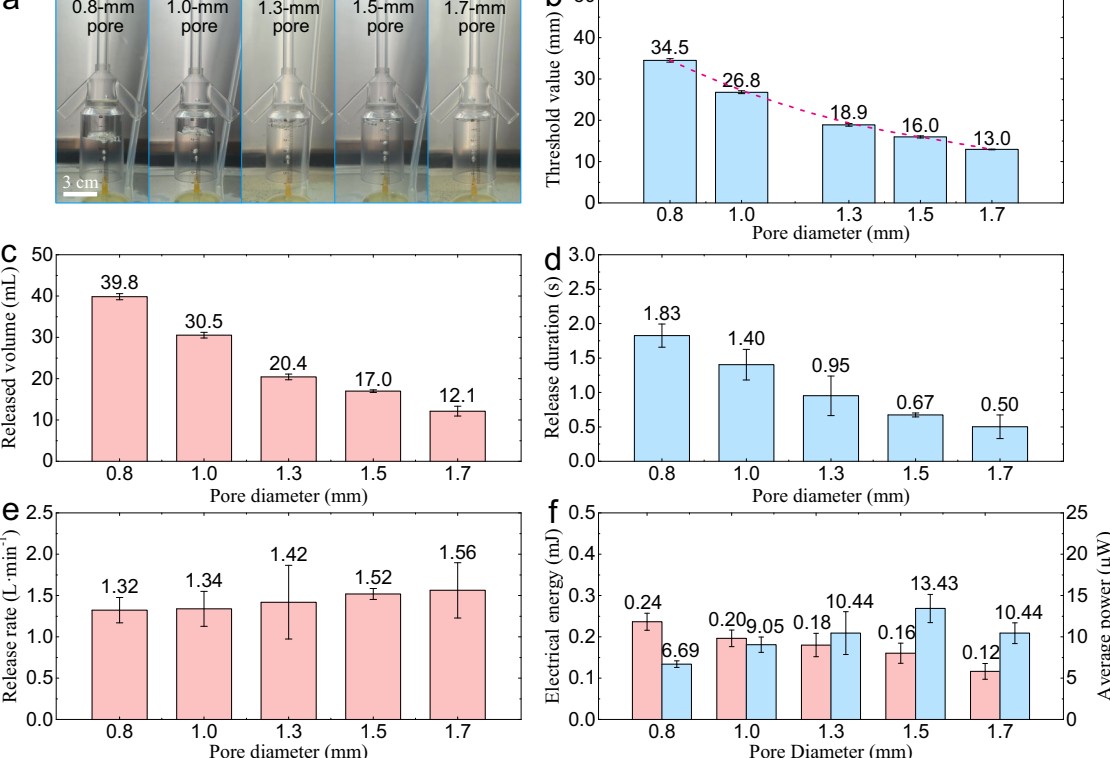

**Fig. 4 | Threshold characteristics of the porous valves corresponding to the porous plates with different pore diameters.** All error bars in this figure represent the standard deviation. **a** Photographs of the gas accumulated in the porous valves with the pore diameters of 0.8 mm, 1.0 mm, 1.3 mm, 1.5 mm, and 1.7 mm, respectively, when the porous valves reach the opening thresholds. **b** Opening thresholds of porous valves with different pore diameters. **c** Mean volume of gas released after each open event. **d** Mean duration of each bubble-release stage. **e** Gas-release rate of the bubble-release stage in each cycle. **f** Average values of the output electrical energy (pink column) and power (light blue column) in each cycle of the BEHD (bubble energy harvesting device) equipped with the porous valves with different pore diameters within 300 s at a gas flux of 70.1 mL·min⁻¹. Source data are provided as a Source data file.

and a high gas-release rate. These experimental phenomena verify the correctness of the theoretical principle. Relying on the liquid surface tension and the specially designed porous interface mechanical structure, the porous valve realizes the self-adaptive dynamic variation of the Laplace pressure difference, increasing first and then decreasing. The dynamic equilibrium of interfacial forces sustains the continuous accumulation of bubbles when the accumulated gas is below the threshold; conversely, once the accumulated gas exceeds the threshold, the positive mechanical feedback effect enables the bubbles to be sufficiently released at a high speed within a short period of time.

Equation (11) in the theoretical analysis indicates that the opening threshold of the porous valve is inversely proportional to the diameter $d_u$ of the upper port of each micropore. To further investigate the threshold characteristics of the porous valve, five types of porous plates with different pore diameters were fabricated and installed inside the porous valve to test the threshold characteristics of the porous valve. The five types of porous plates in the experiment had the same number and arrangement of the micropores, with the only difference in the sizes of the pore diameters.

The photographs of the porous valves (which correspond to porous plates with different pore diameters) are shown in Fig. 4a, captured just as the accumulated gas reaches the threshold. As shown in Supplementary Movie 3, each cycle of the porous valves consists of a bubble-accumulation stage and a high-speed bubble-release stage. The mean values and standard deviations of the opening thresholds of the porous valves with different pore diameters in 8 cycles are listed in Fig. 4b. The mean values of the opening thresholds of the porous valves with the pore diameters of 0.8, 1.0, 1.3, 1.5, and 1.7 mm are 35.4, 26.8, 18.9, 16.0 and 13.0 mm, respectively, corresponding to the

standard deviations of 0.43, 0.34, 0.29, 0.27 and 0.09 mm, respectively. The photographs in Fig. 4a and the experimental results in Fig. 4b show that: (1) The opening threshold of the porous valve decreases as the pore diameter increases, which verifies the results of the theoretical analysis (Eq. (11)). (2) The porous valves corresponding to the porous plates with different pore diameters all have stable opening thresholds.

The decreasing tendency of the opening threshold as the pore diameter increases means that the gas-storage capacity of the porous valve decreases with the pore diameter. As shown in Fig. 4c, the porous valve with a larger pore diameter has a weaker gas-storage capacity (lower maximum gas-storage volume), and thus the volume of gas released after each open event is smaller. Similar results are found in Fig. 4d, where the duration of each bubble-release stage decreases as the pore diameter increases. According to the gas-release volume and duration, the gas-release rate of the bubble-release stage in each cycle is calculated and listed in Fig. 4e. Within the pore diameter range of 0.8–1.7 mm, the mean value of the gas-release rate of the porous valve during the bubble-release stage shows a gradually increasing trend as the pore diameter increases. When the pore diameter exceeds 1.7 mm, as shown in Fig. S8 of the Supplementary information, the volume of gas that the porous valve can accumulate is too small to read the opening threshold, and the bubble-accumulation capacity almost disappears. When the pore diameter is below 0.8 mm, processing errors significantly reduce the uniformity of pore diameters among different micropores on the porous plate. The accumulated gas can only be slowly released from the individual micropore with the largest actual pore diameter on the porous plate surface, leading to a significant decrease in the gas-release rate and making it difficult to

achieve high-speed release of the accumulated gas. In practical applications, if it is necessary to further enhance the gas-storage capacity of the porous valve without reducing the pore diameter, it can be achieved by increasing the cross-sectional area of the gas reservoir.

The mean values of the gas-release volume (Fig. 4c) and the gas-release rate (Fig. 4e) of the porous valve in each cycle exhibit opposite variation trends. Appropriate increases in both contribute to improving the flow rate of the fluid in the bubble rising pipe and the output power of the BEHD. Therefore, it is difficult to directly judge the impact of the pore diameter on the output performance of the device. To further investigate the impact of the pore diameter, the electrical energy output by the BEHD assembled with the porous valves of different pore diameters was tested by experiment. At a gas flux of 70.1 mL· min$^{-1}$, the current and electrical energy output by the device at its optimum external resistance (35 Ω) are plotted in Fig. S9 in the Supplementary information. During bubble accumulation, the pipe contains neither bubbles nor upwelling flows, and the turbine generator remains motionless without current output. Once in the bubble-release stage, the work performed by bubble buoyancy releases buoyancy potential energy, driving the pipe's fluid and the turbine to make the device generate current.

As shown in Fig. S9a–e, during a 300-s observation period, the porous valves with the pore diameters of 0.8, 1.0, 1.3, 1.5, and 1.7 mm release bubbles at a high speed for 8, 14, 17, 25, and 26 times, respectively. Each bubble-release event is accompanied by a single current output from the BEHD. The maximum output currents of the devices corresponding to the porous valves with the five pore diameters are 4.47, 4.23, 3.30, 3.08, and 2.43 mA, respectively. The average values of the electrical energy that the devices output in each cycle are 0.24 mJ, 0.20 mJ, 0.18 mJ, 0.16 mJ, and 0.12 mJ, respectively, as illustrated in Fig. 4f. A porous valve with a larger pore diameter has a smaller volume of gas released in each cycle, resulting in less gas buoyancy potential energy being input into the energy harvesting device in each bubble-release process. Therefore, both the maximum output current and the electrical energy generated in each cycle decrease as the pore diameter increases. However, it does not mean that a porous valve with a larger pore diameter has more advantages in bubble energy harvesting. Over 300 s, the average output powers of the BEHD assembled with the porous valves of different pore diameters in each cycle are 6.69 μW, 9.05 μW, 10.44 μW, 13.43 μW, and 10.44 μW, respectively, exhibiting a tendency of increasing first and then decreasing (Fig. 4f). Similar results are found in Fig. S9f (Supplementary information). Over 300 s, the electrical energy generations of the devices corresponding to the porous valves with different pore diameters are 1.89 mJ, 2.75 mJ, 3.06 mJ, 4.01 mJ, and 3.03 mJ, respectively, also showing a trend of increasing first and then decreasing. Among the five energy harvesting devices, the one assembled with the 1.5-mm pore diameter porous valve exhibits the highest output power and total electrical energy, thus demonstrating greater advantages in bubble energy harvesting. This is the result of the combined effect of the gas-release volume and the gas-release rate in each cycle. For a porous valve with a small pore diameter, the low gas-release rate and the small number of times of bubble release led to the low output power of the device. When the pore diameter of a porous valve is too large, the released gas has a small volume and weak buoyancy. This, in turn, lowers the fluid flow rate in the bubble rising pipe and the turbine's rotation rate, thereby decreasing the electrical output.

The above experimental results illustrate that the opening threshold of the porous valve decreases as the pore diameter increases, and the porous valves with different pore diameters all have stable opening thresholds. The porous valve with a higher threshold has a stronger gas-storage capacity, a larger gas-release volume, and a longer duration of gas released in each cycle, but a lower gas-release rate during the opening stage of the porous valve. By appropriately increasing the pore diameter of the porous plate to enhance the gas-

release rate, it is possible to not only solve the problem of the low gas-release rate for the porous valve with a high threshold, but also avoid the problems of the small gas-release volume in each cycle and the low output power caused by an overly large pore diameter.

## Bubble energy harvesting performance

To further validate the advantages of the porous valve based on the passive porous interface mechanical structure for harvesting energy from low-flux bubbles, we compared the energy harvesting performance of two identical devices: one equipped with a 1.5-mm pore diameter porous valve, and the other without any valve. As illustrated in Fig. 5b, in the comparative experimental setup, bubbles below the valveless device enter the bubble rising pipe directly without prior accumulation.

At different environmental gas fluxes, the ratio of the gas release rate (the gas-intake rate of the bubble rising pipe) of the BEHD equipped with a porous valve to that of the BEHD with no valve is shown in Fig. 5a. According to the experimental results in Fig. 4e, the gas-release rate of the porous valve with a pore diameter of 1.5 mm is 1.517 L· min$^{-1}$ during the bubble-release stage. Unlike the BEHD equipped with the porous valve, the gas-intake rate of the device with no valve is equal to the environmental gas flux. The results in Fig. 5a prove that the introduction of the porous valve significantly enhances the gas-release rate, especially at low gas fluxes. When the environmental gas flux is as low as 11.0 mL· min$^{-1}$, the ratio of the gas-release rate of the porous valve during the bubble-release stage to that of the device with no valve reaches 138. By virtue of the introduction of the porous valve, the instantaneous gas intake rate of the energy harvesting device has been significantly increased.

The increase in the instantaneous gas-release rate has changed the flow pattern of the gas-liquid two-phase flow in the bubble rising pipe. As shown in Fig. 5b and Supplementary Movie 4, bubbles entering the pipe of the valveless device remain in a discrete state, because this device has a low gas-intake rate and the bubbles' diameters are much smaller than the pipe's inner diameter. The discrete bubbles rise slowly in the pipe, forming a low-speed bubble flow. Unlike the valveless BEHD, a large amount of gas accumulated under the porous valve-based BEHD is released at a high speed through the porous plate in less than 1 s. Within a short time, these gases rush into the BEHD, forming a long baculiform bubble that is sufficient to fill the pipe's inner diameter, with its upper end taking a bullet-like shape. These phenomena indicate that the regulating effect of the porous valve not only increases the instantaneous gas-intake rate of the bubble rising pipe, but also changes the flow pattern of the gas-liquid two-phase flow in the pipe, specifically, from the low-speed bubble flow to the high-speed slug flow.

The change in the flow pattern has significantly enhanced the flow rate of the gas-liquid two-phase flow in the bubble rising pipe, thereby increasing the turbine generator's rotation rate. As presented in Fig. 5c, under a gas flux condition of 70.1 mL· min$^{-1}$, the rotation rate of the turbine driven by low-speed bubble flow in the valveless device consistently fluctuates within a low-speed range (67–77 rpm). In contrast, in the device equipped with the porous valve, the maximum rotation rate of the turbine driven by the high-speed slug flow reaches 500 rpm, which is 6.5 times that of the turbine in the device with no valve. There are two main reasons for this result. Firstly, the gas volume fraction of the slug flow is much higher than that of the bubble flow. A large volume of gas can provide greater buoyancy as a driving force, thereby significantly increasing the flow rate of the gas-liquid two-phase flow in the pipe. Moreover, the rising process of a single bubble sufficient to fill the pipe diameter avoids the local vortices and internal friction of the fluid generated by the discrete bubbles in the bubble flow, thus further reducing the loss of fluid kinetic energy.

The high gas-release rate, elevated fluid flow rate, and increased turbine rotation rate collectively contribute to enhancing the energy

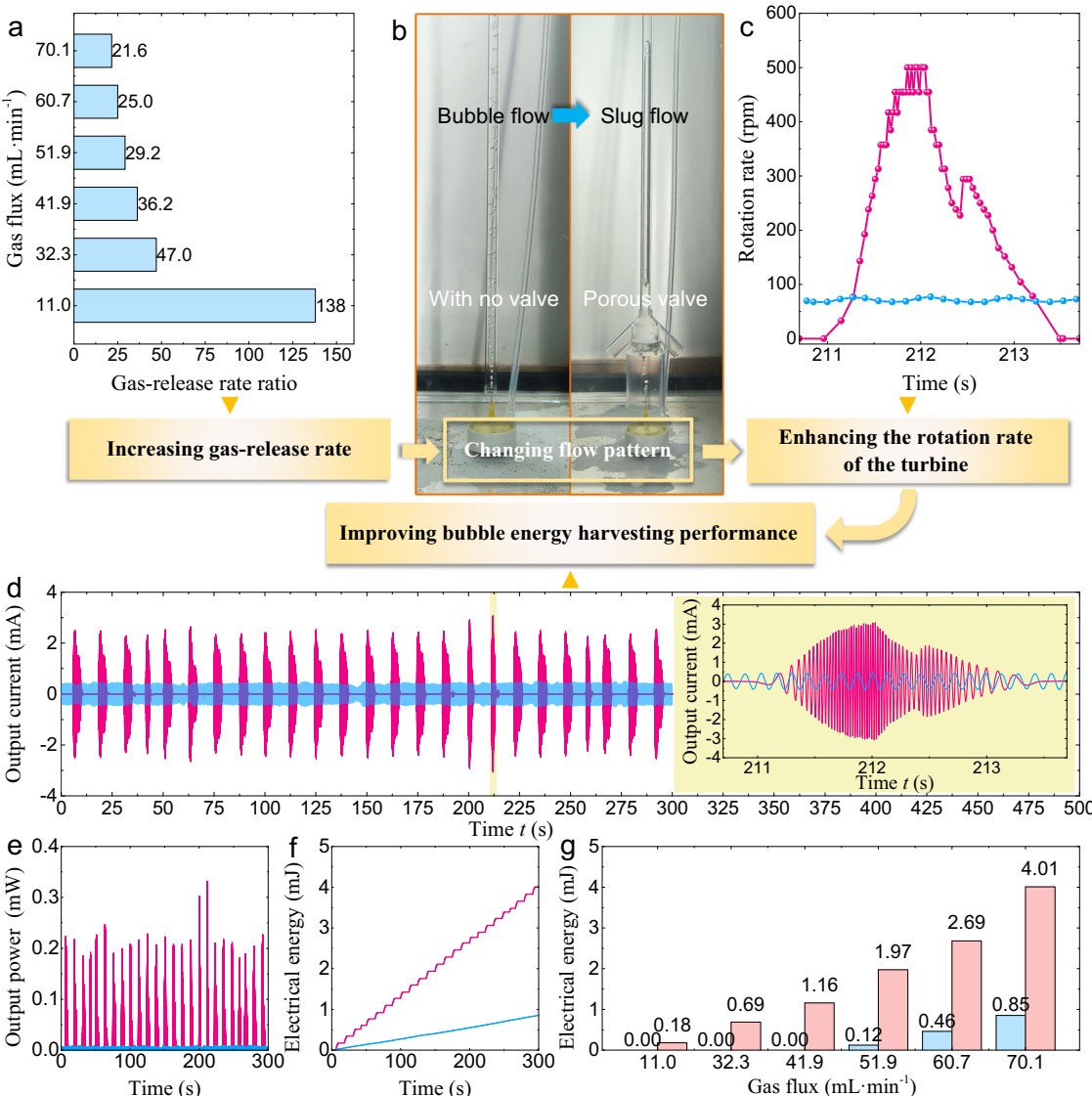

**Fig. 5 | Comparison between the BEHD (bubble energy harvesting device) equipped with the porous valve and the same BEHD with no valve. a** Ratio of the gas-release rates of the BEHD equipped with the porous valve to that of the BEHD with no valve. **b** Variation of bubbles and flow patterns in the BEHD at a gas flux of 70.1 mL· min⁻¹. **c** Rotation rate of the turbine generators in the BEHD with a porous valve (pink line) and the BEHD with no valve (light blue line) during the bubble rising process (210.7–213.7 s). **d** Current, **e** power, and **f** electrical energy output by

the BEHD to an external resistor of 35 Ω. The pink lines represent the output of the BEHD equipped with a porous valve, while the light blue lines represent the output of the BEHD with no valve. **g** Electrical energy output by the BEHD equipped with a porous valve (pink column) and the BEHD with no valve (light blue column) over 300 s at different environmental gas fluxes. Source data are provided as a Source data file.

harvesting performance of the BEHD under low gas flux conditions. The current that the energy harvesting device outputs to its optimal load resistance (35 Ω) is presented in Fig. 5d. For the BEHD with no valve, the amplitude of its output current is basically stable, with a maximum value of only 0.51 mA. In contrast, for the BEHD equipped with the porous valve, the amplitude of its output current exhibits visible periodic variation. The output current exhibits 25 peaks over 300 s, which correspond to 25 cycles of bubble accumulation and high-speed release by the porous valve. For example, during the period from 204 to 211 s, the porous valve is in the bubble-accumulation stage, with no bubbles present in the BEHD. The liquid is in a state of rest, and thus the output current of the turbine generator is zero. At 211 s, the gas released by the porous valve enters the BEHD at a high speed. The bubble buoyancy does positive work, generating a rapidly rising slug flow and converting the buoyancy potential energy into the kinetic energy of the slug flow. The fluid drives the turbine and transfers

energy to it. Consequently, the rotation rate of the turbine generator quickly increases from 0 to 500 rpm within 0.9 s (Fig. 5c), and the corresponding amplitude of the output current rapidly increases from 0 to the maximum value of 3.08 mA (inset in Fig. 5d). When the long baculiform bubble enters the turbine, the turbine is filled with gas, and thus the driving force of the fluid acting on the turbine decreases instantaneously, resulting in a significant reduction of the output current from 3.08 mA at 212 s to 1.39 mA at 212.4 s. When the bubbles completely leave the turbine, the local low pressure at the tail of the bubbles reduces the resistance acting on the liquid below the turbine and generates a high-speed wake flow. The wake flow transfers energy to the turbine again, accelerating its rotation rate from 227 to 294 rpm within 0.1 s. As a result, the turbine generator experiences a brief increase in the amplitude of its output current by 0.48 mA within 0.1 s, from 1.39 to 1.87 mA. With the wake flow disappearing, the flow rate of the fluid inside the pipe exhibits a gradual reduction, which

subsequently induces a corresponding decline in the rotation rate and output current of the turbine generator; both parameters decrease to zero at 213.5 s. By virtue of the slug flow generated by the high-speed release of bubbles, the peak output current of the porous valve-equipped BEHD reaches 6.04 times that of the valveless BEHD. The comparison results in Fig. 5e, f demonstrate that the BEHD integrated with a porous valve exhibits a maximum power of 0.33 mW and a total electrical energy output of 4.01 mJ over 300 s. These parameters are roughly 36.6 and 4.7 times that of the valveless counterpart, which has a maximum power of 9.01 μW and a 300-s electrical energy output of $8.54 \times 10^{-1}$ mJ. The introduction of the porous valve significantly improves the energy harvesting performance of low-flux bubbles.

To further validate the advantages of the porous valve in low-flux bubble energy harvesting, Fig. 5g provides comparative data on the electrical energy output of two BEHDs, one with a porous valve and the other without, under different gas flux conditions. At low gas fluxes of 11.0, 32.3, and 41.9 mL·min⁻¹, the turbine generator in the BEHD with no valve cannot be driven (Supplementary Movie 4), producing no electrical output. This is because the hydrodynamic force provided by the low-speed gas-liquid two-phase flow is lower than the start-up threshold of the turbine generator. In contrast, the BEHD based on the porous valve outputs 0.18, 0.69, and 1.16 mJ of electrical energy over 300 s at the low gas fluxes of 11.0, 32.3, and 41.9 mL·min⁻¹, respectively. By virtue of the favorable capabilities of bubble accumulation and high-speed release, the instantaneous gas-intake rate of the BEHD based on the porous valve has been increased by 1–2 orders of magnitude. Under low-gas-flux conditions, high-speed slug flow is still generated and drives the turbine generator to rotate rapidly and output electrical energy, thus overcoming the limitations of low gas flux. When the gas flux surpasses 41.9 mL·min⁻¹, the output performance of the porous valve-equipped BEHD continues to be substantially higher than that of the valveless BEHD. For example, at the gas flux of 51.9 mL·min⁻¹, the turbine generator of the valveless BEHD rotates at a low speed and often pauses (Fig. S10 in the Supplementary information), and the electrical energy production over 300 s is only 0.12 mJ. For the BEHD equipped with the porous valve, the electrical energy production is as high as 1.97 mJ under the same conditions, which is 16.4 times that of the BEHD with no valve. These experimental phenomena and results prove that the introduction of the porous valve enables the energy harvesting device to overcome its threshold barrier for starting operation and has significant advantages in underwater energy harvesting from low-flux bubbles, while the device without a valve lacks operational capability in underwater environments characterized by low gas flux. In addition, by virtue of the stable opening threshold (Fig. 4b) of the porous valve, as shown in Fig. S10f in the Supplementary information, the number of times of bubble release by the porous valve over 300 s is approximately proportional to the ambient gas flux. Each time of gas release corresponds to one start-up and the current output of the BEHD. Therefore, the operation frequency of the BEHD based on the porous valve can reflect the magnitude of the ambient gas flux. Besides being used for energy harvesting, the porous valve-equipped BEHD is expected to be used as a passive sensing device for submarine gas flux.

The BEHD assembled with a porous valve outperforms the valveless one in maximum output power and electrical energy production, with respective increases of 36.6 and 16.4 times. The increase in maximum output power is primarily attributed to the enhancement in the instantaneous gas intake rate of the device. The significant improvement in electrical energy production arises from that the energy harvesting scheme of bubble accumulation and high-speed release significantly increases the rotational speed-flux coefficient λ of the BEHD. According to Eq. (S8) in the Supplementary information, the output energy density (electrical energy obtained from a unit volume of bubbles) of the BEHD is proportional to the square of the turbine rotational speed $n_t$ and inversely proportional to the gas-intake flux $Q_g$

of the bubble rising pipe. Increasing the rotational speed-flux coefficient $\lambda = n_t^2/Q_g$ of the BEHD contributes to enhancing the output energy density. The experimental results in Table S1 indicate that when the gas-intake flux of the BEHD increases, the electrical energy harvested from unit volume of bubbles also increases. Therefore, if the total volume $V_g$ of bubbles entering the BEHD is constant, increasing the gas-intake rate by shortening the gas-intake duration can enhance the electrical energy production. For example, at a gas flux of 51.9 mL·min⁻¹, the bubble accumulation and high-speed release behavior of the porous valve shortens the gas-intake duration from 300 s for the valveless device to 57.3 s for the porous valve device. The gas-intake flux and average turbine rotational speed increase from 51.9 mL·min⁻¹ and 22.4 rpm for the valveless device to 272 mL·min⁻¹ and 158 rpm for the porous valve device, respectively. Correspondingly, the rotational speed-flux coefficient λ increases from $1.61 \times 10^5$ to $1.53 \times 10^6$ r² m⁻³ s⁻¹, representing an increase of nearly an order of magnitude. Thus, the bubble energy harvesting efficiency of the device is significantly enhanced.

Table 1 provides a comparison between the porous valve-based BEHD and existing devices in terms of application scenarios, energy conversion principles, and output energy density. Among them, the diving-surfacing device driven by chemical reaction relies on the buoyancy of carbon dioxide bubbles generated by calcite dissolution in acidic solutions to drive a wire to move in a magnetic field, thus being suitable for acidic solution environments. Its energy conversion efficiency from chemical energy to electrical energy is $1.5 \times 10^{-5}$%[49]. The bubble energy harvesters based on electrostatic induction currently apply only to freshwater environments due to the problem of charge decay in saline solutions. The pipe fluid energy harvesting device driven by bubble buoyancy used in this work has no charge decay problem and is suitable for operation in seawater environments. According to the experimental results shown in Fig. 5f, at a gas flux of 70.1 mL·min⁻¹, the BEHD with no valve generates $8.54 \times 10^{-1}$ mJ of electrical energy from 350.5 mL bubbles over 300 s. The corresponding energy density and energy harvesting efficiency (Section 10 in the Supplementary Information) are 2.44 mJ·L⁻¹ and 0.03%, respectively. Under the same conditions, the porous valve-equipped BEHD outputs 4.01 mJ of electrical energy. The corresponding energy density and energy harvesting efficiency are 11.44 mJ·L⁻¹ and 0.14%, respectively, both of which are 4.7 times those of the valveless BEHD. The output energy density of the porous valve-based BEHD has reached 6.7 times that of the existing bubble energy generator[36], and the energy harvesting efficiency of the porous valve-based device is four orders of magnitude higher than that of the diving-surfacing device driven by chemical reaction. These comparative results confirm that the BEHD equipped with the porous valve achieves favorable energy harvesting performance and maintains high operational efficiency.

## Applications

The porous valve-based BEHD can effectively collect energy from underwater low-flux bubbles, thereby overcoming the limitation of low gas flux in the actual underwater environment, such as photosynthetic bubbles generated by aquatic plants or the bubbles released by submarine methane seepage. To further evaluate the practicability of the device, low-flux oxygen bubbles generated by the photosynthesis of Riccia fluitans were used as the object for energy harvesting in the following demonstration experiment of bubble energy harvesting and applications. The electrical energy generated by the device undergoes processing via an energy harvesting circuit, after which it is stored in a capacitor with a capacitance of 470 μF.

Figure 6a shows the accumulation process of low-flux oxygen bubbles released by the underwater Riccia fluitans through photosynthesis into the gas reservoir of the porous valve. Upon initial formation, bubbles are minuscule in volume, with buoyancy too weak to overcome adhesion forces, thus adhering to the surface of Riccia

**Table 1 | Comparison between existing researches and the BEHD (bubble energy harvesting device) equipped with the proposed MSPV (mechanical self-adaptive porous valve)**

| Applicable scenarios | References | Energy conversion principle | Volume of bubbles $V_g$ (mL) | Gas flux $Q_g$ (mL·min$^{-1}$) | Electrical energy output $E_e$ (mJ) | Energy density $E_d$ (mJ·L$^{-1}$) |
|---|---|---|---|---|---|---|
| Acidic solution | Diving-surfacing device driven by chemical reaction[49] | Chemical reactions and electromagnetic induction | 176 | 35.2 | $1.8 \times 10^{-5}$ | $1.02 \times 10^{-1}$ |
| Fresh water | Fluidic electricity generator[51] | Electrostatic induction | 1.14 | 30 | $9.38 \times 10^{-7}$ | $8.23 \times 10^{-4}$ |
| | Dual-electrode electrets[34] | Electrostatic induction | 0.189 | 10 | $7.0 \times 10^{-5}$ | $3.70 \times 10^{-1}$ |
| | Multi-electrode electrets[35] | Electrostatic induction | 0.25 | 10 | $3.9 \times 10^{-5}$ | $1.56 \times 10^{-1}$ |
| | Bubble energy generator[36] | Electrostatic induction | 0.1 | 10 | $1.72 \times 10^{-4}$ | 1.72 |
| | Lubricant-impregnated bubble energy generator[37] | Electrostatic induction | 0.1 | 10 | $1.06 \times 10^{-4}$ | 1.06 |
| Seawater | Broadband rotary hybrid generator[18] | Electromagnetic induction and piezoelectric effect | 1500 | / | 2.15 | 1.43 |
| | Biomimetic cactus[48] | Electromagnetic induction | 18.7 | 16.4 | $1.17 \times 10^{-1}$ | 6.26 |
| | BEHD without the MSPV | Electromagnetic induction | 350.5 | 70.1 | $8.54 \times 10^{-1}$ | 2.44 |
| | BEHD with the MSPV | Electromagnetic induction | 350.5 | 70.1 | 4.01 | 11.44 |

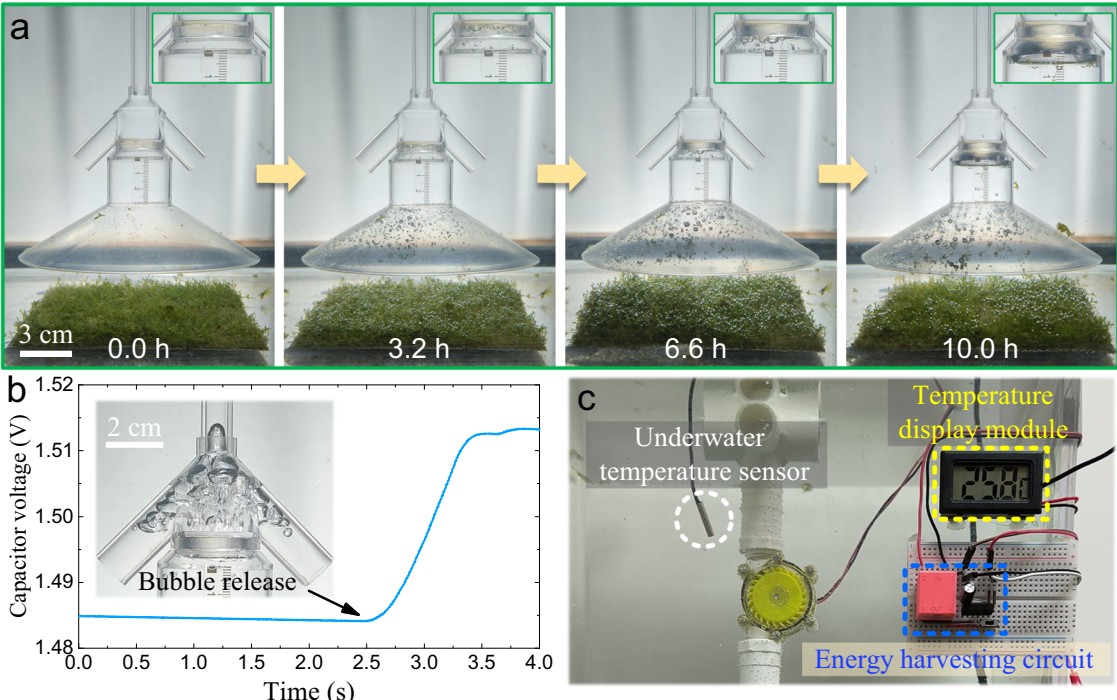

**Fig. 6 | Harvesting and utilization of the low-flux bubble energy generated by photosynthesis. a** Production and accumulation process of low-flux photosynthetic bubbles below the porous valve. **b** Voltage of the energy-storage capacitor before and after bubble release. **c** The collected bubble energy powers an underwater temperature sensor with temperature display module. Source data are provided as a Source data file.

fluitans. As photosynthesis proceeds, the attached bubbles gradually expand. When their diameter reaches the millimeter scale, buoyancy overcomes adhesion, prompting detachment of bubbles from the plant surface. In the experiment, the released photosynthetic bubbles have a diameter of approximately 1–3 mm. After 10 h of accumulation, approximately 17 mL of gas have accumulated in the gas reservoir of the porous valve. The corresponding average gas flux in the underwater environment is about 28 μL·min$^{-1}$. As shown in the Supplementary Movie 5, the microbubbles released by the Riccia fluitans gradually enter the gas reservoir of the porous valve. When the liquid-level difference corresponding to the accumulated gas reaches the opening threshold of the porous valve, the bubbles are immediately

released through the porous interface mechanical structure in less than 1 s (inset in Fig. 6b) and then flow into the bubble rising pipe. It means that the gas intake rate of the pipe has been increased by four orders of magnitude compared to the environmental gas flux. In the course of bubble ascent, the buoyancy potential energy possessed by the released gas is initially converted into fluid energy. This fluid energy is thereafter converted into electrical energy by the BEHD, with the resulting electrical energy stored in the capacitor. As shown in Fig. 6b, before one event of bubble release, the voltage of the capacitor is 1.484 V. The capacitor is charged as the bubbles rise, of which the voltage gradually increases to 1.513 V and stops increasing after the generator stops rotating. As shown in Fig. 6c, when the energy-storage

capacitor is connected to an underwater temperature sensor with a rated voltage of 1.5 V, the sensor is immediately driven and its temperature display module shows that the current water temperature is 25.8 °C. It means that the energy of the low-flux bubbles produced by photosynthesis has been collected and successfully used to drive the underwater sensing device. These experimental results verify that the BEHD utilizing a porous valve has achieved effective harvesting of bubble potential energy at a low gas flux of 28 μL· min$^{-1}$. It is expected to be widely used in the subsea observation network to provide in-situ energy for sensors for submarine environment monitoring.

## Discussion

This paper proposes a passive porous interface mechanical structure with both self-adaptive mechanical effects and high gas permeability. A mechanical self-adaptive porous valve based on the porous structure was fabricated for efficient energy harvesting from underwater low-flux bubbles. Unlike traditional mechanical metamaterials that rely on external forces to generate their own structural deformation, the proposed passive mechanical structure achieves fluid control by utilizing the deformation of the gas-liquid interface instead of its own structural deformation, and thus does not consume energy. Mechanical analysis shows that, relying on the effect of surface tension, each convex gas-liquid interface inside the conical micropore array of the hydrophilically treated porous structure generates a Laplace pressure difference that varies adaptively in response to the liquid pressure difference. Threshold characteristic experiments verify that the Laplace pressure difference increases in synchronization with the liquid pressure difference. This process suppresses bubble release and facilitates the continuous accumulation of low-flux bubbles until the porous valve attains its opening threshold. After the porous valve reaches the threshold, the Laplace pressure difference decreases rapidly and promotes the high-speed release of the accumulated gas from numerous micropores in a positive feedback manner. The porous valves have stable opening thresholds inversely proportional to the pore diameters. An appropriate increase in the pore diameter of the porous structure contributes to enhancing the gas-release rate. Bubble energy harvesting experiments prove that the instantaneous gas-intake rate of the porous valve-based BEHD is as high as 1.517 L· min$^{-1}$, which is two orders of magnitude higher than that of the device with no valve and four orders of magnitude higher than the gas flux (28 μL· min$^{-1}$) of photosynthetic bubbles generated by aquatic plants. The high gas-intake rate enables the flow pattern in the bubble rising pipe to change from low-speed bubble flow to high-speed slug flow and increases the rotation rate of the turbine generator to 6.5 times that of the device without a valve. The high flow rate and high rotation rate enhance the maximum output power and electrical energy production of the device by factors of 36.6 and 16.4, respectively, compared with the device with no valve. The energy density of bubble energy harvesting reaches 11.44 mJ· L$^{-1}$, which is 6.7 times that of the existing bubble energy generator. Notably, through theoretical derivation of the energy conversion process combined with quantitative experimental research, this paper also reveals the fundamental physical mechanism by which bubble accumulation and high-speed release improve energy harvesting efficiency. Specifically, the physical process of accumulation and release increases the rotational speed-flux coefficient of the BEHD-a key physical parameter proportional to the output energy density of the device. The porous valve-based BEHD achieves a one-order-of-magnitude increase in the rotational speed-flux coefficient compared to the valveless BEHD, thus significantly improving the energy harvesting performance.

The proposed mechanical self-adaptive porous valve has a stable opening threshold inversely proportional to its pore diameter, and the opening threshold is proven to be independent of the number of micropores. Therefore, increasing the number and density of micropores in the structure not only does not affect the bubble accumulation capacity, but also contributes to enhancing the gas permeability. By virtue of the high gas permeability of the passive porous interface mechanical structure, the porous valve achieves a gas-release rate of 1.517 L· min$^{-1}$, which is 7.7 times the maximum gas-release rate (197 mL· min$^{-1}$) of the biconical tube structure[31] in our group's previously proposed biomimetic cactus[48]. In the future, the gas-release rate can be further enhanced by further increasing the number and density of micropores in the structure. Moreover, in the bubble buoyancy-driven pipe fluid energy harvesting systems, bubble rising pipes with different diameters and different turbine generators may require matching gas-liquid two-phase flows with different volume fractions of gas phase to achieve efficient actuation. Thus, prior to practical application, threshold matching tests should be conducted in advance. Rationally designing the opening threshold of the porous valve to adjust the gas-release volume per cycle contributes to further improving the rotational speed-flux coefficient and maximizing the bubble energy harvesting efficiency. Optimizing the gas-release volume per cycle by rationally designing the opening threshold of the porous valve contributes to further improving the rotational speed-flux coefficient and maximizing the bubble energy harvesting efficiency. Indeed, the mechanical self-adaptive porous valve serves as a universal passive bubble control device. Beyond the pipe fluid energy harvesting system used in this study, electrostatic and triboelectric bubble energy harvesters can similarly enhance energy harvesting efficiency by replacing bubble flow with slug flow to achieve rapid movement of bubbles along the surface of electret or triboelectric materials[42]. Therefore, the proposed mechanical self-adaptive porous valve is also expected to be applied to other types of bubble energy harvesting systems, such as electrostatic and triboelectric systems, enhancing bubble energy harvesting performance by bubble accumulation and high-speed release.

In the demonstration experiment of photosynthetic bubble energy harvesting, the gas reservoir of the porous valve collects bubbles released by aquatic plant photosynthesis through a funnel structure, achieving accumulation of bubble buoyancy potential energy. For future practical applications, considering various disturbance factors in submarine environment, such as ocean currents, tide fluctuations, marine organism movement, and subsea geological activities, microbubbles are prone to random dispersive motions. Photosynthetic bubbles released by different aquatic plants may vary in size. Under external disturbances, bubbles of different sizes exhibit differing degrees of random dispersive motions. These factors affect gas collection efficiency. To further enhance the environmental adaptability of the BEHD, a high-efficiency capture device for multidirectional dispersed microbubbles can be developed and installed beneath the porous valve. It contributes to improving the capture efficiency of microbubbles and the gas intake of energy harvesting devices, thereby boosting the power generation capacity at the source. In addition, considering potential fouling in marine environments, biomimetic principles can be adopted to design self-cleaning surface topography and microstructures based on the passive porous interface mechanical structure in the porous valve, further enhancing the durability of the porous valve in marine environments.

Bubble accumulation and high-speed release enable the porous valve-based BEHD to overcome the threshold barrier for starting operation. The device not only effectively harvests the energy from low-flux bubbles represented by the photosynthetic bubbles released by aquatic plants, but also is expected to be used as a passive underwater gas flux detection device. The power generation potential of the BEHD can be further estimated based on the research data of global marine carbon budgets. Seaweed forests in the coastal vegetated ecosystems absorb 56 Tg of carbon annually through photosynthesis and export organic carbon across the continental shelf[50]. This photosynthetic process corresponds to an estimated annual oxygen production of approximately $4.9 \times 10^{12}$ L (under the seawater pressure

conditions at a depth of 200 m). Based on the output energy density (11.44 mJ·L$^{-1}$) of the bubble energy harvesting experimental device over a 0.8-m bubble ascent height from Table 1 and Eq. (S18) in the Supplementary information, the annual electricity generation potential from the photosynthetic bubbles over a 200-m ascent height is calculated to reach a substantial value of 12.4 GWh. The proposed passive porous interface mechanical structure relying on surface tension opens up a passive design philosophy for mechanical materials and devices, provides a promising solution for efficient energy harvesting from low-flux bubbles widely distributed in subsea environments, and is expected to provide reliable in-situ energy sources for subsea distributed self-powered sensing and deep-sea exploration.

## Methods

### Fabrication of the porous valve

As presented in Fig. S1 of the Supplementary Information, the porous valve is assembled from three components, namely, the gas-liquid mixing chamber located at the top, the passive porous plate positioned in the middle, and the gas reservoir situated at the bottom. All three components are manufactured using 3D printing technology with photopolymer resin as the matrix material, adopting the photocuring molding method. The porous plate is constructed from white resin, while the gas-liquid mixing chamber and gas reservoir are fabricated using transparent resin. This transparent design enables clear observation of the processes involved in bubble accumulation and release. The lower rim of the gas-liquid mixing chamber matches the upper rim of the gas reservoir, and the porous plate is embedded between them. The porous plate is a circular thin plate with a diameter of 30 mm and a thickness of 1.5 mm, and it is perforated by numerous conical micropores. The angle between the generatrix of each conical micropore and the lower surface of the porous plate is 60°. The upper end of the generatrix is a 30° arc with a radius of 1 mm.

### Experimental methods

In the threshold characteristic experiment and the bubble energy harvesting experiment, the energy harvesting device was immersed in an aqueous solution with a salinity of 3.5% prepared from sea salt and tap water to simulate the seawater environment. In the experiments corresponding to Figs. 3–6 and Figs. S7–S12 in the Supplementary Information, the porous valve was positioned at the lower end of the bubble rising pipe, with its installation details depicted in Fig. S4 of the Supplementary Information. The working principle of the experimental setup is introduced in Section 3 of the Supplementary information. In the experiment corresponding to Fig. 3, the diameter of the upper edge of the micropores on the porous plate installed inside the porous valve is 0.8 mm. The photos in Fig. 3 and Supplementary Movie 1 were recorded at a speed of 1069 frames per second using a high-speed camera (Model Chronos 1.4, supplied by Kron Technologies Inc.) with a microscope lens. In the experiment corresponding to Fig. 4, the conical micropores of different porous plates have the same generatrix, and the only difference is the size of the pore diameter. The pore diameters marked in Fig. 4a are the diameters of the upper edges of the micropores on the porous plates. The opening threshold of the porous valve can be adjusted by replacing porous plates with different pore diameters. To read the liquid level position at the moment accumulated gas reaches the threshold, a transparent flexible scale was affixed to the outer wall of the gas reservoir beneath the porous plate. The bubble energy harvesting experiment shown in Fig. 5 follows the common approach of comparative experiments in this field[31,48]. In the experiment shown in Fig. 6, four pieces of 7.5 × 7.5 cm$^2$ of Riccia fluitans were placed at the bottom of the water tank, producing oxygen through photosynthesis and providing low-flux bubbles for the energy harvesting device. An LED light source (SL100D, GODOX Photo Equipment Co., Ltd) with a color temperature of 5600K was placed outside the water tank to provide continuous illumination for the

Riccia fluitans. The gas reservoir of the porous valve collected the photosynthetic bubbles released by the Riccia fluitans through a funnel structure with a diameter of 16 cm. The output of the turbine generator in the BEHD was connected to an energy-storage capacitor after being boosted by a 40-fold transformer and rectified by a full bridge.

### Data measurement and calculation

In the experiment of Fig. 4b, the scale value corresponding to the liquid level in the gas reservoir when the porous valve reached the opening threshold was recorded by shooting a video. The opening threshold was calculated by adding the vertical distance (11.5 mm) between the upper surface of the porous plate and the zero scale line of the gas reservoir to the scale value corresponding to the liquid level in the gas reservoir. The data in Fig. 4b–e were all calculated based on the statistical results obtained from eight individual bubble accumulation and release events for the porous valves with different pore diameters. In Fig. 4c, the volume of bubbles released in each cycle is determined by summing two components: the reduced volume of gas within the gas reservoir and the gas intake into the gas reservoir during the bubble-release stage. The gas-release rate in Fig. 4e was calculated by dividing the volume of gas released by the porous valve in each cycle by the corresponding gas-release duration. The rotation rate of the turbine generator depicted in Fig. 5b was computed on the basis of the output current frequency. For the measurement and recording of the turbine generator's output, a digital multimeter (Model DMM6001, supplied by Guangzhou Zhiyuan Electronics Co., Ltd) was employed. The output power data depicted in Fig. 5d was derived from the formula $P(t) = I^2(t)R_{ex}$. In this formula, $I(t)$ represents the instantaneous output current, $R_{ex}$ denotes the resistance value of the load resistor (specifically 35 Ω, provided by the resistance box presented in Fig. S4b of the Supplementary information), and $t$ indicates the time variable. The electrical energy production of the device in Figs. 4f, 5f, S9, and S10 was derived using the formula $E(t) = \int_0^t I^2(t)R_{ex}dt$. The gas fluxes on the abscissa in Fig. 5a, g were derived from the results in Fig. S5 of the Supplementary Information. The results in Fig. 5g were calculated based on the data in Fig. S10. For the energy density presented in Table 1, its calculation formula is $E_d = E_e/V_g$[31,48]. In this formula, $V_g$ refers to the total volume of gas that is employed to produce electrical energy $E_e$. The voltage of the capacitor in Fig. 6b was measured using a digital multimeter (Model DMM6500, supplied by Tektronix, Inc.).

## Data availability

The experimental data generated in this study are provided in the Source Data file and in Figshare (https://doi.org/10.6084/m9.figshare.30166948). Source data are provided with this paper.

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

## Acknowledgements

This work was financially supported by the National Natural Science Foundation of China (Grant No. 62403095, Y.D.), the China Postdoctoral

Science Foundation (Grant No. 2024M760314, Y.D.), the Fundamental Research Funds for the Central Universities (Grant No. 3132025109, Y.D.), the Key Program for Basic Research of China (Grant No. JCKY2023206B026, Y.F.), the National Key Research and Development Program of China (Grant No. 2022YFB4301401, Y.F.), and the Pilot Base Construction and Pilot Verification Plan Program of Liaoning Province of China (Grant No. 2022JH24/10200029, Y.F.).

## Author contributions

Y.D. and P.L. conceptualized the study. Y.D. designed the prototype, conducted the experiments, analyzed the data, and wrote the original draft. P.L., Y.W., Y.F., and Z.L. participated in scientific discussions, reviewed, and edited the manuscript. All the authors reviewed and approved the final manuscript.

## Competing interests

The authors declare no competing interests.
