## [Transparent Peer Review file · Nature Communications]

Mechanical self-adaptive porous valve relying on surface tension for efficient energy harvesting from ultralow-flux bubbles

Corresponding Author: Dr Yu Du

Version 0:

Reviewer comments:

Reviewer #1

(Remarks to the Author)

The manuscript "Passive porous interface-mechanical metamaterial relying on surface tension for efficient energy harvesting from ultralow-flux bubbles" presents a passive porous interface valve with self-adaptive mechanical properties designed to enhance energy harvesting performance. This is achieved by controlling bubble accumulation and enabling high-speed release to improve energy extraction from ultralow-flux bubbles. The authors propose a passive valve design that leverages gas-liquid interface deformation to regulate the bubble gas-intake rate, thereby generating a high-speed fluid capable of producing sufficient force to drive power generation in a bubble energy harvesting system. The article is well-structured, providing an in-depth analysis while maintaining a balance between fundamental principles and technical details.

The following suggestions should be addressed to further improve the quality of the manuscript:

1. The proposed valve design is referred to as a "mechanical metamaterial" in the manuscript. However, mechanical metamaterials are typically defined as rationally designed artificial structures that exhibit unusual physical and mechanical properties. While the valve design is structurally engineered, it does not appear to demonstrate unexpected or unusual behavior. Furthermore, similar structures have been previously utilized as valve in gas-liquid interface studies. Therefore, it may not be appropriate to classify the proposed structure as a metamaterial. A more suitable term could be "mechanical self-adaptive porous valve." The authors are encouraged to reconsider this terminology.
2. There is an unusual hyphen between "interface" and "mechanical" in the title and the phrase "interface-mechanical" within the text. The authors should either remove the hyphen or clarify its intended meaning.
3. In the section Fabrication of the Porous Valve, the statement "The angle between the generatrix of ... a radius of 1 mm" raises a question regarding the selection of the arc angle for each conical micropore (reported as 60° with respect to the porous plate). How was this angle determined? Would varying the tapering angle not affect the gas-liquid interface dynamics? Additionally, while the outer diameter (or radius) of the cone varied five times, other parameters were kept fixed. Was this choice based on a specific theoretical basis? The authors should provide further clarification.
4. Please include discussion on the limitations of proposed energy harvesting approach or future work. This could include details on the design and characteristics of porous valve, and other different parameters.
5. The manuscript should be carefully reviewed for any grammatical errors or typographical mistakes, and corrections should be made as necessary.

Reviewer #2

(Remarks to the Author)

I have thoroughly reviewed the manuscript "Passive porous interface-mechanical metamaterial relying on surface tension for efficient energy harvesting from ultralow-flux bubbles" submitted for consideration in the Journal Nature Communications. The authors report a design that combines a bubble energy harvester with a porous valve that allows for bubble accumulation and high-speed release, leading to a higher electrical output compared to previous works. The general scope of this work is written and systematically done. Still, there are some questions that could be asked about the presented

manuscript, suggesting further explanation and detail should be made. Therefore, I recommend the publication of this work in Nature Communications if the following comments can be fully addressed.

1. The authors need to clarify the function of extra water inlets on both sides of the device. How does it influence the intern flow of the gas-liquid two-phase flow?
2. The current study investigates bubble dynamics (generation, accumulation, and release) within stable liquid systems characterized by strictly controlled unidirectional bottom-to-top water flow. However, to validate the system's practical viability for oceanic energy harvesting, it is imperative to conduct additional stability analyses under dynamic marine conditions that are associated with turbulent flows, multidirectional currents, and wave-induced perturbations.
3. The discussion on the performance improvement from Structure 1 (with no valve) to Structure 2 (with porous valve) is unconvincing. Considering that the input energy is the total kinetic energy of bubbles, the authors should explain why bubble accumulation and high-speed release can improve the average power output. Formula derivation and further quantitative investigation are preferred.
4. The data in Fig. 4e does not match the description: "According to the gas-release volume and duration, the gas-release rate of the bubble-release stage in each cycle is calculated and listed in Figure 4e.". The data in Fig.4c and Fig 4d appear mathematically insufficient to reconstruct the reported trend in Fig. 4e. The authors may use the larger difference in pore diameters to support their claim.
5. The rotational dynamics in Figs. 5c and 5d exhibit an anomalous double-peak pattern within individual cycles, particularly evident between 212s and 213s where the turbine's rotational velocity unexpectedly increases. What is the reason for this secondary acceleration of turbine rotation?
6. What is the total efficiency of energy transfer? How does it compare with other designs? As far as I am concerned, there are many different types of bubble generators. The author should make a comparison with other types of bubble generators, not only the metamaterial methods.
7. The Demo in Fig. 6 did not have any scale bar. Based on the size of the device, I guess the bubbles generated by photosynthesis are quite tiny. The authors should discuss the effect of bubble size on the system.
8. Life-span performance should be considered. Making allowance for the long-term use inside the seawater, the erosion of bubbles and water flow on devices, especially on the hydrophilic coatings, cannot be ignored.

Reviewer #3

(Remarks to the Author)

The manuscript titled "Passive porous interface-mechanical metamaterial relying on surface tension for efficient energy harvesting from ultralow-flux bubbles" presents a passive method that utilizes surface-tension-driven gas-liquid interface deformation to harvest energy from ultralow-flux underwater bubbles. Unlike conventional active systems, the design operates without external energy input, enabling self-adaptive bubble accumulation and release via Laplace pressure modulation. A porous valve based on this concept improves gas intake and energy conversion efficiency, with experimental results demonstrating effective energy harvesting from biologically generated bubbles. The study combines theoretical analysis and experimental validation, highlighting the potential of this approach for electrical energy production in support of subsea self-powered sensing and in situ autonomous exploration. Overall, the concept is interesting and can lead to a practical device.

Here are a few comments and questions:

1. In the abstract, "The maximum output power and electrical energy production are enhanced by factors of 36.63 and 16.42, respectively". Please clarify what baseline this enhancement is compared to in the abstract.
2. On page 6, the porous interface-mechanical metamaterial is treated with a hydrophilic modification. Could the authors comment on the long-term durability of the air-dried nano-silica coating under seawater conditions? In addition, beyond the porous material itself, was any surface treatment applied to the internal surface of the reservoir structure? As observed in Figure 6, numerous bubbles appear to be pinned on the inner wall of the reservoir inlet (the flared, funnel-shaped structure), which may reduce the efficiency of bubble collection.
2. Based on the introduction (page 1), the observed gas fluxes in actual marine environments are on the order of 6.75 mL/min and 2.18 mL/min, which are lower than the experimental condition of 70.1 mL/min (page 9). Could the authors provide an estimate of the energy that could be generated under real-world conditions using this passive porous interface-mechanical metamaterial over a period of operation?
3. What is the effective range of pore sizes that allows the proposed concept to function reliably? As the pore size decreases, is there a risk that the accumulated gas pressure could displace the liquid back into the bulk rather than allowing the gas to pass through the porous interface?
4. On page 9, the authors state that Figure 4e indicates an increasing trend in gas-release rate with increasing pore diameter. However, the error bars shown in Figure 4e suggest that the differences in gas-release rate between pore sizes in the range of 1.3 to 1.7 mm may not be significant, making the overall trend less clear. Could the authors clarify whether the observed trend can be considered general across all experimental conditions?
5. On page 12, Table 1 lists the gas flux used in this work as 70.1 mL/min. If applicable, could the authors include the gas flux values used in other referenced works in the table for better comparison?

Reviewer #4

(Remarks to the Author)

I co-reviewed this manuscript with one of the reviewers who provided the listed reports. This is part of the Nature

Communications initiative to facilitate training in peer review and to provide appropriate recognition for Early Career Researchers who co-review manuscripts.

Version 1:

Reviewer comments:

Reviewer #1

(Remarks to the Author)

I can affirm that all my concerns have been satisfactorily addressed; the manuscript can be accepted in its current form.

Reviewer #2

(Remarks to the Author)

The authors have addressed all my concerns and I suggest the publication of this manuscript in Nature Communications.

Reviewer #3

(Remarks to the Author)

Dear Reviewers,

First of all, I would like to appreciate for your review efforts and critiques on our manuscript. According to your very useful comments, we have revised this paper with a significant amount of efforts. The valuable comments from the reviewers have helped us to improve the quality and completeness of this paper significantly. We have revised the manuscript to the best of our ability. However, we would be open to revising further should any further suggestions from the reviewers arise. Thank you for your time and effort on this paper.

We have highlighted all the changes in the revised manuscript. The major differences from previous version are summarized as follows.

- (1) We have revised the title of this paper to "Mechanical self-adaptive porous valve relying on surface tension for efficient energy harvesting from ultralow-flux bubbles".
- (2) The principle of the bubble energy harvesting device based on the porous valve is further explained in Section 3 of the Supplementary Information. Table 1 in the main text has been supplemented with a more comprehensive comparison between the proposed device and various existing bubble energy harvesters.
- (3) The mechanism by which bubble accumulation and high-speed release behaviors enhance the device's average output power is analyzed through theoretical derivation combined with quantitative experiments, with the relevant content added to Section 6 in the Supplementary Information.
- (4) The influence of the cone half-angle of micro-pores on the threshold characteristics of the porous valve is discussed via theoretical analysis and experiments, and the corresponding results are added to Section 7 in the Supplementary Information.
- (5) Further experiments on the threshold characteristics of porous valves with different pore diameters have been conducted, and the related content is added to Section 8 in the Supplementary Information.
- (6) The stability of the electrical output of the bubble energy harvesting device under external turbulent disturbances has been tested. The stability of the aerophobic characteristics of the passive porous interface mechanical structure under bubble plume aeration impingement has been evaluated. Relevant contents are added to Section 12 in the Supplementary Information.

We have addressed all the points raised by the reviewers, as summarized below.

Reviewer #1:

The manuscript "Passive porous interface-mechanical metamaterial relying on surface tension for efficient energy harvesting from ultralow-flux bubbles" presents a passive porous interface valve with self-adaptive mechanical properties designed to enhance energy harvesting performance. This is achieved by controlling bubble accumulation and enabling high-speed release to improve energy extraction from ultralow-flux bubbles. The authors propose a passive valve design that leverages gas-liquid interface deformation to regulate the bubble gas-intake rate, thereby generating a high-

speed fluid capable of producing sufficient force to drive power generation in a bubble energy harvesting system. The article is well-structured, providing an in-depth analysis while maintaining a balance between fundamental principles and technical details.

The following suggestions should be addressed to further improve the quality of the manuscript:

1. The proposed valve design is referred to as a "mechanical metamaterial" in the manuscript. However, mechanical metamaterials are typically defined as rationally designed artificial structures that exhibit unusual physical and mechanical properties. While the valve design is structurally engineered, it does not appear to demonstrate unexpected or unusual behavior. Furthermore, similar structures have been previously utilized as valve in gas-liquid interface studies. Therefore, it may not be appropriate to classify the proposed structure as a metamaterial. A more suitable term could be "mechanical self-adaptive porous valve." The authors are encouraged to reconsider this terminology.

Reply: Thank you for your valuable suggestion. We have revised the terminology in the title and main text of our manuscript according to your suggestion. The "mechanical self-adaptive porous valve" is established as the primary subject of this study, and the original term "passive porous interface-mechanical metamaterial" in the manuscript has been modified to "passive porous interface mechanical structure". As you noted, the porous structure in the proposed porous valve differs significantly from traditional mechanical metamaterials. The proposed passive porous interface mechanical structure relies on surface tension to control bubble accumulation and release, thus requiring no additional energy consumption. Therefore, classifying the proposed structure as a metamaterial may be inappropriate. Compared with "passive porous interface-mechanical metamaterial", "passive porous interface mechanical structure" is a more accurate description. The "mechanical self-adaptive porous valve" based on this "passive porous interface mechanical structure" is the core content of this paper. The revised title is "Mechanical self-adaptive porous valve relying on surface tension for efficient energy harvesting from ultralow-flux bubbles", which better aligns with the theme of this manuscript.

2. There is an unusual hyphen between "interface" and "mechanical" in the title and the phrase "interface-mechanical" within the text. The authors should either remove the hyphen or clarify its intended meaning.

Reply: Thank you for your valuable suggestion. We have removed the hyphen between "interface" and "mechanical" in the manuscript. Previously, we used the hyphen to indicate that the mechanical behavior of the proposed passive porous interface mechanical structure occurs at the interface. We found that removing the hyphen does not affect the intended meaning.

3. In the section Fabrication of the Porous Valve, the statement "The angle between the generatrix of ... a radius of 1 mm" raises a question regarding the selection of the arc angle for each conical micropore (reported as 60° with respect to the porous plate). How was this angle determined?

Would varying the tapering angle not affect the gas-liquid interface dynamics? Additionally, while the outer diameter (or radius) of the cone varied five times, other parameters were kept fixed. Was this choice based on a specific theoretical basis? The authors should provide further clarification.

Reply: Thank you for your valuable suggestion. This paper proposes a mechanical self-adaptive porous valve that enables passive control of automatic accumulation and high-speed release of ultralow-flux bubbles. When the accumulated gas reaches the opening threshold of the porous valve, it marks the turning point where the porous valve transitions from the bubble-accumulation stage to the bubble-release stage. Therefore, the opening threshold is the most crucial characteristic parameter of the porous valve, which determines the mechanical behavior of the gas-liquid interfaces bubble accumulation capacity of the porous valve. Accordingly, this paper conducts a detailed experimental study on the threshold characteristics of the mechanical self-adaptive porous valve in the section "Threshold characteristic". According to Equation (11) in the theoretical analysis of the main text, the opening threshold of the porous valve is $\Delta h_T = 4\gamma / \rho g d_u$, indicating that the diameter d_u of the upper port of each micropore is the only structural parameter determining the opening threshold. Based on this theoretical foundation, the experimental research focuses on investigating the influence of pore size on the threshold characteristics of the porous valve.

Regarding the cone half-angle of the micropore, since it is theoretically irrelevant to the opening threshold of the porous valve, this parameter is not discussed in the main text. To further improve the completeness of this paper, we have included the theoretical analyses of the micropore cone half-angle in Section 7 of the Supplementary Information. Additionally, we have fabricated a group of porous plates with the same upper port diameter d_u (1.5 mm) but different cone half-angles and conducted a comparative experiment to investigate the cone half-angle, aiming to validate the theoretical analyses. The experimental results have further verified that the opening threshold of the porous valve is independent of the cone half-angle.

The supplemental discussion in Section 7 of the Supplementary Information is as follow.

7. Discussion about the cone half-angle of the micropores

7.1 Bubble-accumulation stage

During the bubble-accumulation stage, the interfacial forces acting on the gas in the mechanical self-adaptive porous valve maintain dynamic equilibrium. The Laplace pressure difference $\Delta p = 2\gamma/R$ across the curved gas-liquid interface within the micropores is equal in magnitude to the liquid pressure difference $p_d - p_u = \rho g \Delta h$ between the upper and lower gas-liquid interfaces of the gas, from which it follows that

$$R = \frac{2\gamma}{\rho g \Delta h}, \quad (S9)$$

where R is the radius of curvature of the gas-liquid interface, γ represents the surface tension of the

liquid, ρ denotes the liquid density, g is the acceleration of gravity, and Δh is the liquid-level difference between the upper and lower gas-liquid interfaces of the gas. As shown in Figure S6, the angle between the generatrix of the micropore and the vertical axis (cone half-angle) is α . The angle between the curved gas-liquid interface and the solid-liquid interface is θ . The three-phase contact line between the gas-liquid interface and the inner wall of the micropore is a circle with radius r . According to geometric relations, it can be derived that

$$R = \frac{r}{\cos(\alpha + \theta)} \quad (\text{S10})$$

The functional relationship between the radius r of the three-phase contact line and the liquid-level difference Δh can be derived by substituting Equation (S10) into Equation (S9), as

$$r = \frac{2\gamma \cos(\alpha + \theta)}{\rho g \Delta h} \quad (\text{S11})$$

The variation in the radius r of the three-phase contact line reflects the positional variation of the gas-liquid interface within the micropore, where a decrease in r indicates that the interface moves upward. According to Equation (S11), as the liquid level difference Δh gradually increases, the radius r of the three-phase contact line adaptively decreases. The corresponding result is that the gas-liquid interface and the three-phase contact line in the micropore adjust their positions adaptively and move upward. During the upward movement of the gas-liquid interface, r and $\alpha + \theta$ decrease. According to Equation (S10), the curvature radius R of the gas-liquid interface consequently decreases and gradually approaches the radius r of the three-phase contact line.

Figure S6 Mechanical analysis of the gas-liquid interface at the micropores during the process of bubble accumulation and release.

The above theoretical analysis indicates that during the bubble-accumulation stage, the volume of gas accumulated in the porous valve determines the magnitude of the liquid level difference Δh ,

and Δh determines the liquid pressure difference $\rho g \Delta h$ exerted on the gas. To maintain dynamic mechanical equilibrium, the gas-liquid interface in the micropore adaptively adjusts its curvature radius R , thereby generating a Laplace pressure difference $2\gamma/R$ that balances the liquid pressure difference $\rho g \Delta h$. According to Equation (S9), the curvature radius R of the gas-liquid interface is completely determined by the liquid level difference Δh and is independent of the cone half-angle α of the micropore. Therefore, α does not affect the curvature radius R of the gas-liquid interface and the Laplace pressure difference $2\gamma/R$. Moreover, considering that the radius r of the three-phase contact line and the curvature radius R of the gas-liquid interface are related by the geometric Equation (S10), the radius r of the three-phase contact line is determined not only by the curvature radius R but also influenced by the cone half-angle α of the micropore. It means that the position of the gas-liquid interface depends not only on Δh but also on α . Therefore, changing the cone half-angle α of the micropores on the porous plate can only alters the specific position of the curved gas-liquid interface within the micropores during the process of bubble accumulation, without affecting the curvature radius of the gas-liquid interface and the Laplace pressure difference.

7.2 Bubble-release stage

As shown in Figure S6, when the three-phase contact line coincides with the upper port of the micropore and the gas-liquid interface takes a hemispherical shape, the curvature radius R of the gas-liquid interface reaches the minimum value, which is equal to the radius r_u of the upper port of the micropore ($R_{\min} = r = r_u$), and is independent of the cone half-angle α of the micropore. At this moment, the accumulated gas reaches the opening threshold Δh_T of the porous valve

$$\Delta h_T = \frac{2\gamma}{\rho g r_u} \quad (\text{S12})$$

The porous valve starts to enter the bubble-release stage.

During the bubble-release stage, as shown in Figure S6, the bubbles on the upper surface of the porous plate expand rapidly, and the three-phase contact line expands quickly toward the periphery of the micropore, causing the radius r of the three-phase contact line to increase rapidly. According to the geometric relationship, the curvature radius of the gas-liquid interface is

$$R = \frac{r}{\sin \theta} \quad (\text{S13})$$

In this stage, the Laplace pressure difference across the gas-liquid interface is

$$\Delta p = \frac{2\gamma \sin \theta}{r}, \quad (\text{S14})$$

which is determined by the angle θ between the gas-liquid interface and the solid-liquid interface and the radius r of the three-phase contact line, but independent of the cone half-angle α of the micropore.

Based on the above theoretical analyses, both in the bubble-accumulation stage and the bubble-

release stage, the curvature radius R of the gas-liquid interface, the Laplace pressure difference Δp , and the opening threshold Δh_T of the porous valve are all independent of the cone half-angle α of the micropore. The influence of the cone half-angle α is only reflected in changing the specific position of the curved gas-liquid interface in the micropore during the bubble-accumulation stage, without affecting the opening threshold of the porous valve and the processes of bubble accumulation and release.

7.3 Comparative experiment with different cone half-angles

To further verify the results of the theoretical analysis, four porous plates with different cone half-angles were fabricated and installed inside the porous valve to experimentally test the threshold characteristics of the porous valve. In the experiment, all four porous plates have the same micropore upper port diameter $d_u = 1.5$ mm, with identical quantity and arrangement of micropores. The only difference among the four porous plates is the cone half-angle α of the micropores.

Figure S7 Experimental results of the comparison of opening thresholds for the porous valves with different cone half-angles.

As shown in Figure S7, the cone half-angles of the micropores on the four porous valves are 15°, 30°, 45°, and 60°, respectively. When the gas accumulated in the four porous valves reaches the thresholds, the scales corresponding to the gas-liquid interface below the gas are all between 4 mm and 5 mm. This indicates that the opening thresholds of all porous valves fall within the range of 15.5–16.5 mm, which is close to the experimental result of 16.0 mm for the porous valve with a 1.5-mm pore diameter in Figures 4(a) and (b) of the main text. The experimental results further confirm that the threshold characteristics of the porous valve are independent of the cone half-angle of the micropores.

4. Please include discussion on the limitations of proposed energy harvesting approach or future work. This could include details on the design and characteristics of porous valve, and other different

parameters.

Reply: Thank you for your valuable suggestion. We have supplemented several aspects for further improvement and optimization of the porous valve in future practical applications in the section "Discussion" of the main text, with the specific contents as follows.

The proposed mechanical self-adaptive porous valve has a stable opening threshold inversely proportional to the size of its pore diameter. In the bubble buoyancy-driven pipe fluid energy harvesting systems, bubble rising pipe with different diameters and different types of turbine generators may require matching gas-liquid two-phase flows with different volume fractions of gas phase to achieve efficient actuation. Thus, prior to practical application, threshold matching tests should be conducted in advance. Rationally designing the opening threshold of the porous valve to adjust the gas-release volume per cycle contributes to further improving the rotational speed-flux coefficient and maximizing the bubble energy harvesting efficiency. Optimizing the gas-release volume per cycle by rationally designing the opening threshold of the porous valve contributes to further improving the rotational speed-flux coefficient and maximizing the bubble energy harvesting efficiency. Indeed, the mechanical self-adaptive porous valve serves as a universal passive bubble control device. Beyond the pipe fluid energy harvesting system used in this study, electrostatic and triboelectric bubble energy harvesters can similarly enhance energy harvesting efficiency by replacing bubble flow with slug flow to achieve rapid movement of bubbles along the surface of electret or triboelectric materials. Therefore, the proposed mechanical self-adaptive porous valve is also expected to be applied to other types of bubble energy harvesting systems, such as electrostatic and triboelectric systems, enhancing bubble energy harvesting performance by bubble accumulation and high-speed release.

In the demonstration experiment of photosynthetic bubble energy harvesting, the gas reservoir of the porous valve collects bubbles released by aquatic plant photosynthesis through a funnel structure, achieving accumulation of bubble buoyancy potential energy. For future practical applications, considering various disturbance factors in submarine environment, such as ocean currents, tides fluctuations, marine organism movement, and subsea geological activities, microbubbles are prone to random dispersive motions. Photosynthetic bubbles released by different aquatic plants may vary in size. Under external disturbances, bubbles of different sizes exhibit differing degrees of random dispersive motions. These factors affect gas collection efficiency. To further enhance the environmental adaptability of the bubble energy harvesting device, a high-efficiency capture device for multidirectional dispersed microbubbles can be developed and installed beneath the porous valve. It contributes to improve the capture efficiency of microbubbles and the gas intake of energy harvesting devices, thereby boosting the power generation capacity at the source. In addition, considering potential fouling in marine environments, biomimetic principles can be adopted to design self-cleaning surface topography and microstructures based on the passive porous interface mechanical structure in the porous valve, further enhancing the durability of the porous valve in marine environments.

5. The manuscript should be carefully reviewed for any grammatical errors or typographical mistakes, and corrections should be made as necessary.

Reply: Thank you for your suggestion. We have carefully checked and revised the entire manuscript.

Reviewer #2:

I have thoroughly reviewed the manuscript "Passive porous interface-mechanical metamaterial relying on surface tension for efficient energy harvesting from ultralow-flux bubbles" submitted for consideration in the Journal Nature Communications. The authors report a design that combines a bubble energy harvester with a porous valve that allows for bubble accumulation and high-speed release, leading to a higher electrical output compared to previous works. The general scope of this work is written and systematically done. Still, there are some questions that could be asked about the presented manuscript, suggesting further explanation and detail should be made. Therefore, I recommend the publication of this work in Nature Communications if the following comments can be fully addressed.

1. The authors need to clarify the function of extra water inlets on both sides of the device. How does it influence the intern flow of the gas-liquid two-phase flow?

Reply: Thank you for your valuable suggestions and questions. There are water inlets on both sides of the proposed mechanical adaptive porous valve, of which the function is to construct a complete and unobstructed internal-external water circulation loop for the energy harvesting device, thereby ensuring the rapid flow of the gas-liquid two-phase fluid in the pipe. To intuitively explain the function of the water inlets and the fluid motion process inside and outside the device, we have drawn a schematic diagram of the bubble energy harvesting process (Figure S3 in the Supplementary Information) and supplemented relevant descriptions in Section 3 of the Supplementary Information, which are as follows.

3. Principle of bubble energy harvesting

The working principle of the bubble energy harvesting device based on the mechanical self-adaptive porous valve is shown in Figure S3. The gas-liquid mixing chamber of the porous valve, the bubble rising pipe, and the external water environment form a complete internal and external circulation loop of water flow. The water inlets on both sides of the gas-liquid mixing chamber of the porous valve are the openings through which water from the external environment enters the energy harvesting device. The water outlet above the turbine generator is the opening through which water returns from the device to the external water environment. These inlets and outlets are

essential, as they ensure the smooth circulation of the loop and the rapid upwelling of the gas-liquid two-phase fluid in the bubble rising pipe.

Figure S3 Working principle of the bubble energy harvesting device.

The working process of the bubble energy harvesting device is as follows. After the bubbles accumulated in the gas reservoir are released through the porous plate, they start to float upward and enter the bubble rising pipe. The bubbles rise in the pipe to form an upwelling flow, leading to a higher flow rate in the pipe than that in the external water environment (which can be approximated as a quasi-static liquid). According to Bernoulli's principle, the fluid pressure inside the device is much lower than that in the external water environment, thereby generating a pressure difference at the water inlets on both sides of the porous valve. Due to the pressure difference, water from the external water environment is drawn into the pipe through the water inlets. At the same time, the rising gas-liquid two-phase fluid in the pipe flows through the turbine and exits from the outlet at the upper end of the pipe, returning to the external water environment, thus forming a complete water circulation loop. When the fast-rising gas-liquid two-phase flow in the pipe passes through the turbine, it generates a strong driving force to drive the turbine generator to rotate and output electrical energy. In this circulation of water flow, the buoyancy of the bubbles does positive work on the water in the pipe. The buoyancy potential energy of the bubbles is converted into the kinetic energy of the gas-liquid two-phase fluid, which is finally converted into electrical energy by the turbine generator and output to the external load. It should be noted that due to the liquid surface tension, the porous plate inside the porous valve can only allow bubbles to pass through but not water flow. If the water inlets and outlets are not set, the water circulation inside and outside the bubble rising pipe cannot be realized. Accordingly, the bullet-shaped bubbles occupying the entire pipe diameter can only rise slowly within the pipe or even get stuck, leading to blockage. Such

blockage can hinder the formation of upwelling flow, thereby incapacitating the system from supplying adequate fluid power for the rotation of the turbine generator.

2. The current study investigates bubble dynamics (generation, accumulation, and release) within stable liquid systems characterized by strictly controlled unidirectional bottom-to-top water flow. However, to validate the system's practical viability for oceanic energy harvesting, it is imperative to conduct additional stability analyses under dynamic marine conditions that are associated with turbulent flows, multidirectional currents, and wave-induced perturbations.

Reply: Thank you for your valuable suggestions. Employing a pipe to constrain bubble motion is a distinctive characteristic of the bubble buoyancy-driven pipe fluid energy harvesting method proposed by our group, offering two crucial advantages:

(1) Compared with bubbles rising in an open water environment, using a pipe to confine bubbles and liquid for directional flow contributes to reducing energy losses of fluid during bubble ascent, increasing bubble rising velocity, and forming high-speed gas-liquid two-phase flow. This energy harvesting scheme relying on bubble buoyancy to drive the fluid in a pipe has been proven to have high gas-liquid energy transfer and conversion efficiency, as reported in our earlier studies (Reference 22 of this manuscript).

(2) Benefiting from the isolation and shielding effect of the pipe, the flow direction of the fluid inside the pipe always aligns with the pipe axis. The flow state of the fluid in the pipe is less susceptible to disturbance from the complex marine fluid environment, such as the turbulent flows, multidirectional currents, and waves you mentioned, thereby ensuring that the water flow through the turbine generator always moves along the tangential direction of the turbine. Sustained unidirectional flow within the pipe of the energy harvesting device contributes to improve energy conversion efficiency and anti-disturbance capability, which is also an advantage of this energy harvesting method.

To further demonstrate the stability and practicality of the system, we have conducted further experimental tests on the stability of the system by using an underwater propeller to generate multi-directional currents and turbulence outside the pipe and create severe surface waves, simulating complex marine fluid disturbances. The corresponding experimental content and results have been added to Section 12.1 of the Supplementary Information, as detailed below.

12.1 External fluid disturbances

To further demonstrate the ability of the pipe bubble energy harvesting system based on the porous valve to resist external fluid disturbances, the performance of the device under external water flow impact was tested by experiment. As shown in Figure S11(a), an underwater motor-driven propeller was installed on the sidewall of the experimental water tank to generate turbulent disturbances in the external water environment of the bubble energy harvesting device. During the

experiment, the rotation of the underwater propeller vigorously agitated the water around the bubble energy harvesting device, causing a large number of microbubbles to be entrained into the water and follow the turbulent motion. Meanwhile, as shown in Figure S11(b), the rotation of the propeller also caused the water surface to undergo vigorous undulating motion. Under a gas flux of $70.1 \text{ mL} \cdot \text{min}^{-1}$, the motion of the bubble in the bubble rising pipe of the energy harvesting device and the output current of the turbine generator under external turbulent disturbances are presented in Figures S11(c) and (d), respectively.

Figure S11 Stability test of the bubble energy harvesting device under external turbulent disturbances. (a) Turbulent disturbance generated by the rotation of an underwater propeller. (b) Vigorous undulating motion of the water surface. (c) Fluid motion state in the bubble rising pipe of the energy harvesting device under external turbulent disturbances. (d) Output current of the energy harvesting device at a gas flux of $70.1 \text{ mL} \cdot \text{min}^{-1}$.

Despite the intense turbulent disturbances in the external water environment, the bubble energy harvesting device still maintains a stable working state, which is reflected in the motion state of the fluid in the pipe and the electrical output of the device. In terms of the fluid motion state, as shown

in Figure S11(c), during the bubble-release stage, the flow pattern in the pipe remains a stable slug flow, where a long baculiform bubble sufficient to fill the pipe diameter rises rapidly in the pipe. It proves that the bubble rising pipe of the device and the external water environment still maintain the stable water circulation depicted in Figure S3. In terms of electrical output, the peak output current of the device in each bubble-release stage stabilizes at approximately 3 mA, and the number of current peaks within 300 s is 25, which is similar to the electrical output results without external turbulent disturbances shown in Figure 5d of the main text. By virtue of the shielding effect of the bubble rising pipe, the flow direction of the fluid inside the pipe always aligns with the pipe axis and is not disturbed by the complex fluid motion in the external environment, thus ensuring that the water flowing through the turbine generator always moves along the pipe axis (tangential to the turbine). The fluid inside the pipe of the energy harvesting device always maintains a stable state of directional flow and electrical output performance. These experimental results confirm that the bubble energy harvesting system based on the mechanical self-adaptive porous valve exhibits favorable anti-disturbance capability and practical application potential.

Indeed, the motion of ocean currents may affect the trajectory of bubbles in water, thereby influencing the bubble capture efficiency of the energy harvesting device. We are currently investigating a passive method for efficient multi-directional bubble collection and convergence. We invite you to stay tuned for our subsequent research publications.

3. The discussion on the performance improvement from Structure 1 (with no valve) to Structure 2 (with porous valve) is unconvincing. Considering that the input energy is the total kinetic energy of bubbles, the authors should explain why bubble accumulation and high-speed release can improve the average power output. Formula derivation and further quantitative investigation are preferred.

Reply: Thank you for your question. In the previous version of the manuscript, we attributed the increase in output power of the bubble energy harvesting device to the enhancement in the instantaneous gas-intake rate. Indeed, bubble accumulation and high-speed release can significantly enhance the instantaneous gas-intake rate of the bubble rising pipe, which can only effectively explain the reason for the increase in peak output power. To explain the reason for the increase in average output power, we conducted further theoretical derivations, energy conversion process analysis, and quantitative experiments. According to further investigations, we found that the reason for the enhancement of average output power is that bubble accumulation and high-speed release effectively increase the rotational speed-flux coefficient of the bubble energy harvesting device, which is a crucial physical parameter proportional to the output energy density of the device. The theoretical derivations of the energy conversion process have been added to Section 6 of the Supplementary Information, the relevant experimental data have been organized in Table S1 of the Supplementary Information, and the discussion of the experimental results has been supplemented in the seventh paragraph in the section "Bubble energy harvesting performance" of the main text.

Thank you again for your question. Your question has prompted us to deeply reflect on the

fundamental physical principles of using bubble accumulation and high-speed release processes to enhance bubble energy harvesting efficiency, and has further improved the theoretical depth and completeness of this paper.

Section 6 of the Supplementary Information (including Table S1) is as follows.

6. Analysis of the energy conversion process

In the bubble energy harvesting system, as shown in Figure S3, the buoyancy of bubbles drives the fluid circulation inside and outside the device, and the fluid power drives the turbine generator to rotate and output electrical energy. The main function of the system is to convert the buoyancy potential energy of bubbles into electrical energy. The specific energy conversion process is as follows. When bubbles rise in the pipe, the buoyancy of the bubbles does work on the gas-liquid two-phase fluid in the pipe, driving the fluid to flow upward. The buoyancy potential energy E_b of the bubbles is released, with part converted into the kinetic energy E_k of the gas-liquid two-phase fluid, and the rest consumed by overcoming various resistive work, including the viscous resistance loss E_v , the frictional resistance loss E_r from turbine rotation, and the electrical energy E_e generated by overcoming the electromagnetic resistance of the generator. Accordingly, the energy conversion equation of the bubble energy harvesting system is

$$-\Delta E_b = \Delta E_k + E_v + E_r + E_e, \quad (S1)$$

where ΔE_b and ΔE_k represent the variations of the buoyancy potential energy E_b and the kinetic energy E_k , respectively. ΔE_b takes a negative value because the release of buoyancy potential energy occurs as bubbles ascend.

When the rising distance of bubbles is short and the change in hydrostatic pressure acting on the bubbles is insignificant, the variation in bubble volume caused by the pressure change is negligible. During a period t_1 , the amount of buoyancy potential energy released by the bubble is equal to the work done by the buoyancy force, which can be expressed as

$$-\Delta E_b = F_b L = \rho g Q_g t_1 L = \rho g V_g L, \quad (S2)$$

where F_b represents the buoyancy force of bubbles, L represents the length of the vertical pipe, ρ is the liquid density, g is the acceleration of gravity, Q_g is the gas-intake flux of the pipe, and V_g is the total volume of bubbles entering the bubble energy harvesting device over a period t_1 . Driven by the gas-liquid two-phase flow, the turbine generator produces an induced electromotive force of an induced electromotive force of

$$e = NBSk\omega_t \sin k\omega_t t, \quad (S3)$$

where N represents the number of turns in the generator coil, B denotes the magnetic flux density, S is the effective area of the coil, k is the number of magnetic pole pairs, ω_t is the angular velocity of turbine rotation, and t represents time. The resulting loop current is

$$i = \frac{e}{R_{\text{in}} + R_{\text{ex}}} = \frac{NBSk\omega_t \sin k\omega_t t}{R_{\text{in}} + R_{\text{ex}}}, \quad (\text{S4})$$

where R_{in} denotes the internal resistance of the energy harvester, and R_{ex} is the external load resistance. Correspondingly, the output power of the turbine generator is

$$P_{\text{out}} = i^2 R_{\text{ex}} = \frac{R_{\text{ex}} (NBSk\omega_t)^2 \sin^2 k\omega_t t}{(R_{\text{in}} + R_{\text{ex}})^2} \quad (\text{S5})$$

The average output power over the corresponding sine period is

$$P_{\text{ave}} = \frac{R_{\text{ex}} (NBSk\omega_t)^2}{2(R_{\text{in}} + R_{\text{ex}})^2} \quad (\text{S6})$$

When the gas-liquid two-phase fluid in the bubble rising pipe is in a stable flow state, and both the flow rate of the fluid and the rotation rate of the turbine generator remain constant, the electrical energy output by the bubble energy harvesting device over a period t_1 is

$$E_{\text{e}} = P_{\text{ave}} t_1 = \frac{R_{\text{ex}} (NBSk\omega_t)^2 V_{\text{g}}}{2(R_{\text{in}} + R_{\text{ex}})^2 Q_{\text{g}}} \quad (\text{S7})$$

The output energy density of the bubble energy harvesting device (electrical energy obtained from unit volume of bubbles) is

$$E_{\text{d}} = \frac{E_{\text{e}}}{V_{\text{g}}} = \frac{n_{\text{t}}^2}{Q_{\text{g}}} \cdot \frac{2\pi^2 R_{\text{ex}} (NBSk)^2}{(R_{\text{in}} + R_{\text{ex}})^2} = \lambda \cdot \frac{2\pi^2 R_{\text{ex}} (NBSk)^2}{(R_{\text{in}} + R_{\text{ex}})^2}, \quad (\text{S8})$$

where $n_{\text{t}} = \omega_{\text{t}} / (2\pi)$ denotes the rotational speed (number of revolutions per second) of the turbine. Equation (S8) indicates that the output energy density of the bubble energy harvesting device is proportional to the square of the turbine rotational speed n_{t} and inversely proportional to the gas-intake flux Q_{g} of the bubble rising pipe. $\lambda = n_{\text{t}}^2 / Q_{\text{g}}$ can be defined as the rotational speed-flux coefficient. According to Equation (S8), the following conclusion can be drawn: Increasing the rotational speed-flux coefficient of the bubble energy harvesting system contributes to enhancing the output energy density of the bubble energy harvesting device.

To validate the correctness of the above theoretical analysis, quantitative experimental studies were conducted on the rotational speed-flux coefficient and output energy density of the bubble energy harvesting device at different gas-intake fluxes. In this experiment, to facilitate the adjustment and control of the gas-intake flux of the pipe, the bubble energy harvesting device was not equipped with the porous valve, and the bubbles directly entered the bubble rising pipe without accumulation. The data on gas-intake flux are derived from the experimental results in Figure S5(b). The number of revolutions of the turbine is calculated based on the number of cycles of the sinusoidal voltage signal output by the generator to the load resistance (35 Ω). The experimental results are listed in Table S1.

Table S1 Rotational speed-flux coefficient and output energy density at different gas-intake fluxes

Gas-intake flux Q_g (mL·min ⁻¹)	Number of revolutions of the turbine N (r)	Average rotational speed of the turbine n_t (r·s ⁻¹)	Electrical energy output E (mJ)	Rotational speed-flux coefficient λ (r ² ·m ⁻³ ·s ⁻¹)	Energy density E_d (mJ·L ⁻¹)
51.9	112	0.37	0.12	1.61×10^5	0.46
60.7	252	0.84	0.46	6.97×10^5	1.52
70.1	347	1.16	0.85	1.15×10^6	2.43
78.3	416	1.39	1.22	1.47×10^6	3.12
87.1	459	1.53	1.48	1.61×10^6	3.40

As shown in Table S1, both the turbine rotational speed n_t and the rotational speed-flux coefficient λ of the bubble energy harvesting device increase with the increase in the gas-intake flux Q_g of the pipe. For example, when the gas-intake flux of the pipe is 87.1 mL·min⁻¹, the average rotational speed and rotational speed-flux of the energy harvesting device are 1.53 r·s⁻¹ and 1.61×10^6 r²·m⁻³·s⁻¹, respectively, some 4.14 and 10 times that (0.37 r·s⁻¹ and 1.61×10^5 r²·m⁻³·s⁻¹) at the gas-intake flux of 51.9 mL·min⁻¹. Since the rotational speed-flux coefficient λ is proportional to the square of the turbine rotational speed n_t and inversely proportional to the gas-intake flux Q_g , the increase in turbine rotational speed n_t has a more significant effect on enhancing λ than the increase in Q_g . By virtue of the increase in the rotational speed-flux coefficient λ , the output energy density of the device increases from 0.46 mJ·L⁻¹ at 51.9 mL·min⁻¹ to 3.40 mJ·L⁻¹ at 87.1 mL·min⁻¹.

These theoretical analyses and experimental results indicate that when the gas-intake flux of the bubble rising pipe increases, the electrical energy harvested from unit volume of bubbles also increases. In other words, when the total volume V_g of bubbles entering the bubble energy harvesting device is constant, increasing the gas-intake rate of the pipe contributes to enhancing the power generation and average output power of the device. Based on this principle, the mechanical self-adaptive porous valve proposed in this paper is used to accumulate ultralow-flux bubbles and release them into the bubble rising pipe at high speed, thereby significantly increasing the instantaneous gas-intake rate of the pipe. This not only leads to a significant enhancement in the instantaneous output power of the bubble energy harvesting device, but also contributes to the improvement of its average output power.

The supplemented content in the main text is as follows.

The maximum output power and electrical energy production of the bubble energy harvesting device with a porous valve are 36.6 and 16.4 times those of the device with no valve, respectively. The increase in maximum output power is primarily attributed to the enhancement in the instantaneous gas-intake rate of the device. The significant improvement in electrical energy production arises from that the energy harvesting scheme of bubble accumulation and high-speed release significantly increases the rotational speed-flux coefficient λ of the bubble energy harvesting

device. According to Equation (S8) in the Supplementary Information, the output energy density of the bubble energy harvesting device is proportional to the square of the turbine rotational speed n_t and inversely proportional to the gas-intake flux Q_g of the bubble rising pipe. Increasing the rotational speed-flux coefficient $\lambda = n_t^2/Q_g$ of the bubble energy harvesting system contributes to enhancing the output energy density. The experimental results in Table S1 indicate that when the gas-intake flux of the bubble rising pipe increases, the electrical energy harvested from unit volume of bubbles also increases. Therefore, if the total volume V_g of bubbles entering the pipe is constant, increasing the gas-intake rate by shortening the gas-intake duration can enhance the electrical energy production. For example, at a gas flux of $51.9 \text{ mL}\cdot\text{min}^{-1}$, the bubble accumulation and high-speed release behavior of the porous valve shortens the gas-intake duration from 300 s for the valveless device to 57.3 s for the porous valve device. The gas-intake flux and average turbine rotational speed increase from $51.9 \text{ mL}\cdot\text{min}^{-1}$ and 22.4 rpm for the valveless device to $272 \text{ mL}\cdot\text{min}^{-1}$ and 158 rpm for the porous valve device, respectively. Correspondingly, the rotational speed-flux coefficient λ increases from $1.61\times 10^5 \text{ r}^2\cdot\text{m}^{-3}\cdot\text{s}^{-1}$ to $1.53\times 10^6 \text{ r}^2\cdot\text{m}^{-3}\cdot\text{s}^{-1}$, representing an increase of nearly an order of magnitude. Thus, the bubble energy harvesting efficiency of the device is significantly enhanced.

4. The data in Fig. 4e does not match the description: “According to the gas-release volume and duration, the gas-release rate of the bubble-release stage in each cycle is calculated and listed in Figure 4e.”. The data in Fig.4c and Fig 4d appear mathematically insufficient to reconstruct the reported trend in Fig. 4e. The authors may use the larger difference in pore diameters to support their claim.

Reply: Thank you for your suggestion. The data in Figures 4c-e were all calculated based on the results of eight events of bubble accumulation and release using porous valves with different pore diameters. The data in Figure 4e fully correspond to the original data in Figures 4c and 4d. It should be noted that the data in Figure 4e were not directly obtained by dividing the calculated mean values in Figures 4c and 4d. The calculation process for the data in Figure 4e is as follows. We first divided the volume of gas released by the porous valve in each event of bubble release by the corresponding gas-release duration to calculate the gas-release rate in each event. We then calculated the mean value and standard deviation of the gas-release rates of eight events of bubble release, which were presented in Figure 4e. To avoid misunderstanding among readers, we have listed the original data corresponding to Figures 4c-e in Table S2-S4 of the Supplementary Information.

According to the experimental results, within the pore diameter range of 0.8–1.7 mm, the mean value of the gas-release rate of the porous valve during the bubble-release stage shows a gradually increasing trend as the pore diameter increases, but the difference is not particularly significant. Therefore, we use a more rigorous description for Figure 4e in the revised manuscript. In addition, we attempted to expand the range of pore diameter and observed the following experimental phenomena: (1) When the pore diameter of the porous valve exceeds 1.7 mm, the volume of gas that the porous valve can accumulate is too small to read the opening threshold, and the bubble-

accumulation capacity almost disappears. (2) When the pore diameter is below 0.8 mm, processing errors significantly reduce the uniformity of pore diameters among different micropores on the porous plate. The accumulated gas can only be slowly released from the individual micropore with the largest actual pore diameter on the porous plate surface, leading to a significant decrease in the gas-release rate and making it difficult to achieve high-speed release of the accumulated gas.

The seventh paragraph of the section "Threshold characteristic" in the main text has been revised as follows.

The decreasing tendency of the opening threshold as the pore diameter increases means that the gas-storage capacity of the porous valve decreases with the pore diameter. As shown in Figure 4c, the porous valve with a larger pore diameter has a weaker gas-storage capacity (lower maximum gas-storage volume), and thus the volume of gas released after each open event is smaller. Similar results are found in Figure 4d, where the duration of each bubble-release stage decreases as the pore diameter increases. According to the gas-release volume and duration, the gas-release rate of the bubble-release stage in each cycle is calculated and listed in Figure 4e. Within the pore diameter range of 0.8–1.7 mm, the mean value of the gas-release rate of the porous valve during the bubble-release stage shows a gradually increasing trend as the pore diameter increases. When the pore diameter exceeds 1.7 mm, as shown in Figure S8 of the Supplementary Information, the volume of gas that the porous valve can accumulate is too small to read the opening threshold, and the bubble-accumulation capacity almost disappears. When the pore diameter is below 0.8 mm, processing errors significantly reduce the uniformity of pore diameters among different micropores on the porous plate. The accumulated gas can only be slowly released from the individual micropore with the largest actual pore diameter on the porous plate surface, leading to a significant decrease in the gas-release rate and making it difficult to achieve high-speed release of the accumulated gas. In practical applications, if it is necessary to further enhance the gas-storage capacity of the porous valve without reducing the pore diameter, it can be achieved by increasing the cross-sectional area of the gas reservoir.

The corresponding added content in the Supplementary Information is in Section 8.1, as follows.

8.1 Performance of bubble release

The original data corresponding to Figures 4c–e in the main text are listed in Tables S2–S4. Figure S8 shows the photographs of bubble release from the porous valves with pore diameters of 0.7 mm and 1.8 mm.

Table S2 Volume of gas released by the porous valves with different pore diameters after each open event at an environmental gas flux of $70.1 \text{ mL} \cdot \text{min}^{-1}$ (data unit: mL)

Pore diameters d_u (mm)	1st	2nd	3rd	4th	5th	6th	7th	8th	Mean value	Standard deviation
0.8	39.2	38.7	39.9	40.5	41.2	39.9	40.0	39.2	39.8	0.73
1.0	29.9	30.5	32.1	30.2	30.0	30.5	31.1	29.9	30.5	0.69
1.3	20.5	19.7	20.9	21.2	20.9	20.9	20.3	19.0	20.4	0.70
1.5	16.6	17.5	16.4	17.2	17.0	17.1	17.2	17.0	17.0	0.31
1.7	11.2	13.3	10.8	10.9	13.5	13.1	13.5	10.9	12.1	1.20

Table S3 Duration of each bubble-release stage of the porous valves with different pore diameters at an environmental gas flux of $70.1 \text{ mL} \cdot \text{min}^{-1}$ (data unit: s)

Pore diameters d_u (mm)	1st	2nd	3rd	4th	5th	6th	7th	8th	Mean value	Standard deviation
0.8	1.77	1.93	1.85	1.42	2.00	1.83	1.88	1.92	1.83	0.17
1.0	1.67	1.12	1.47	1.60	1.17	1.17	1.70	1.35	1.40	0.22
1.3	1.27	1.10	1.12	0.75	0.68	0.80	1.38	0.52	0.95	0.29
1.5	0.63	0.70	0.70	0.72	0.68	0.63	0.67	0.65	0.67	0.03
1.7	0.35	0.62	0.37	0.35	0.82	0.48	0.68	0.35	0.50	0.17

Table S4 Gas-release rate during the bubble-release stage of the porous valves with different pore diameters in each cycle at an environmental gas flux of $70.1 \text{ mL} \cdot \text{min}^{-1}$ (data unit: $\text{L} \cdot \text{min}^{-1}$)

Pore diameters d_u (mm)	1st	2nd	3rd	4th	5th	6th	7th	8th	Mean value	Standard deviation
0.8	1.33	1.20	1.30	1.71	1.24	1.31	1.27	1.23	1.32	0.15
1.0	1.07	1.64	1.31	1.13	1.54	1.57	1.10	1.33	1.34	0.21
1.3	0.97	1.08	1.12	1.70	1.83	1.57	0.88	2.20	1.42	0.44
1.5	1.57	1.50	1.41	1.44	1.49	1.62	1.54	1.57	1.52	0.07
1.7	1.91	1.29	1.77	1.87	0.99	1.62	1.18	1.87	1.56	0.34

Figure S8: Photographs of bubble-release behavior of the porous valves with pore diameters of 0.7 mm and 1.8 mm. (a) The porous valve with a pore diameter of 0.7 mm releases bubbles from only a single pore. (b) Photograph of the porous valve with a pore diameter of 1.8 mm at the opening threshold, where the gas-liquid interface below the accumulated gas does not reach the scale markings range of the gas reservoir.

In the main text, Figure 4e shows the variation trend of the gas-release rate of the porous valves with pore diameter at an environmental gas flux of $70.1 \text{ mL} \cdot \text{min}^{-1}$. To minimize the influence of the gas-intake behavior on the process of bubble release, eight repetitive experiments were conducted on the porous valves with different pore diameters under the minimum gas flux ($11.0 \text{ mL} \cdot \text{min}^{-1}$) provided by the experimental setup in Figure S4. The corresponding gas-release rates are presented in Table S5. The experimental results show that the average gas-release rate of porous valves during the bubble-release stage still exhibits an increasing trend with the increase in pore diameter.

Table S5 Gas-release rate during the bubble-release stage of the porous valves with different pore diameters in each cycle at a low environmental gas flux of $11.0 \text{ mL} \cdot \text{min}^{-1}$ (data unit: $\text{L} \cdot \text{min}^{-1}$)

Pore diameters d_u (mm)	1st	2nd	3rd	4th	5th	6th	7th	8th	Mean value	Standard deviation
0.8	1.12	1.13	1.20	1.15	1.05	1.25	1.05	1.14	1.14	0.07
1.0	1.19	1.21	1.35	1.13	1.15	1.39	1.47	1.56	1.30	0.15
1.3	1.56	1.43	1.46	1.46	1.36	1.49	1.49	1.58	1.48	0.07
1.5	1.95	2.03	1.66	1.90	1.87	1.95	1.87	1.98	1.90	0.11
1.7	2.26	2.14	2.26	2.06	1.94	2.03	2.19	1.96	2.10	0.12

5. The rotational dynamics in Figs. 5c and 5d exhibit an anomalous double-peak pattern within

individual cycles, particularly evident between 212 s and 213 s where the turbine's rotational velocity unexpectedly increases. What is the reason for this secondary acceleration of turbine rotation?

Reply: Thank you for your question. The reason for the secondary acceleration of turbine is the wake effect of bullet-shaped bubbles. This wake acceleration effect has been observed multiple times in our previous studies (References 22, 23, 24, and 32 of this manuscript). We have further revised the relevant explanation in the fifth paragraph of the section "Bubble energy harvesting performance" in the manuscript, which is as follows.

When the bubbles completely leave the turbine, the local low pressure at the tail of the bubbles reduces the resistance acting on the liquid below the turbine and generates a high-speed wake flow. **The wake flow transfers energy to the turbine again, accelerating its rotation rate from 227 to 294 rpm within 0.1 s.** As a result, the turbine generator experiences a brief increase in the amplitude of its output current by 0.48 mA within 0.1 s, from 1.39 to 1.87 mA.

6. What is the total efficiency of energy transfer? How does it compare with other designs? As far as I am concerned, there are many different types of bubble generators. The author should make a comparison with other types of bubble generators, not only the metamaterial methods.

Reply: Thank you for your question. Based on the experimental data corresponding to Figure 5d, the energy harvesting efficiencies of the bubble energy harvesting device with a porous valve and the device with no valve were calculated to be 0.14% and 0.03%, respectively. This result demonstrates that the porous valve has increased energy harvesting efficiency by a factor of 4.7. The core of this paper is to propose a porous valve that can passively and adaptively control bubble accumulation and high-speed release, aiming to enhance the energy harvesting performance of ultralow-flux bubbles. The results of the comparative experiment effectively demonstrate the advantages of the porous valve. The high-speed slug flow formed by low-flux bubble accumulation significantly improves the energy harvesting efficiency of the device.

It should be noted that the proposed porous valve is not only applicable to the gas-liquid two-phase flow energy harvesting device used in this study, but also suitable for other types of bubble energy harvesters. Existing studies have shown that electrostatic and triboelectric bubble energy harvesters can similarly enhance energy harvesting efficiency by replacing bubble flow with slug flow to achieve rapid movement of bubbles along the surface of electret or triboelectric materials. Therefore, the proposed mechanical self-adaptive porous valve is also suitable for other types of bubble energy harvesting systems, such as electrostatic and triboelectric systems, enhancing bubble energy harvesting performance by bubble accumulation and high-speed release.

According to your suggestions, we have supplemented Table 1 in the main text with a more comprehensive comparison between the gas-liquid two-phase flow energy harvesting device based on the proposed porous valve and various existing bubble energy harvesters. In Section 10 of the

Supplementary Information and the last paragraph of the section "Bubble energy harvesting performance" in the main text, we have added the calculation methods for energy harvesting efficiency and result comparisons, respectively. In addition, we have included future work prospects on efficiency enhancement methods in the "Discussion" section of the main text.

The revised Table 1 is as follow.

Table 1 Comparison between existing researches and the bubble energy harvesting device (BEH) with the proposed mechanical self-adaptive porous valve (MSPV)

Applicable scene	References	Energy conversion principle	Volume of bubbles V_b (mL)	Gas flux Q_g (mL·min ⁻¹)	Electrical energy output E (mJ)	Energy density E_d (mJ·L ⁻¹)
Acidic solution	Diving–surfacing device driven by chemical reaction	Chemical reactions and electromagnetic induction	176	35.2	1.8×10^{-5}	1.02×10^{-1}
Fresh water	Fluidic electricity generator	Electrostatic induction	1.14	30	9.38×10^{-7}	8.23×10^{-4}
	Dual-electrode electrets	Electrostatic induction	0.189	10	7.0×10^{-5}	3.70×10^{-1}
	Multi-electrode electrets	Electrostatic induction	0.25	10	3.9×10^{-5}	1.56×10^{-1}
	Bubble energy generator	Electrostatic induction	0.1	10	1.72×10^{-4}	1.72
	Lubricant-impregnated bubble energy generator	Electrostatic induction	0.1	10	1.06×10^{-4}	1.06
Seawater	Broadband rotary hybrid generator	Electromagnetic induction and piezoelectric effect	1500	/	2.15	1.43
	Biomimetic cactus bubble energy harvester	Electromagnetic induction	18.7	16.4	1.17×10^{-1}	6.26
	BEH without the MSPV	Electromagnetic induction	350.5	70.1	8.54×10^{-1}	2.44
	BEH with the MSPV	Electromagnetic induction	350.5	70.1	4.01	11.44

Section 10 of the Supplementary Information is as follows.

10. Calculation of energy harvesting efficiency

The energy conversion efficiency from bubble buoyancy potential energy E_b to electrical energy E_c is

$$\eta = \frac{E_c}{-\Delta E_b} \times 100\% = \frac{E_c}{\rho g V_g L} \times 100\%, \quad (S15)$$

where E_c represents the generated electrical energy, ΔE_b represent the variation of the buoyancy

potential energy E_b , ρ is the liquid density, g is the acceleration of gravity, V_g is the total volume of bubbles entering the bubble energy harvesting device over a period t_1 , and L represents the length of the vertical bubble rising pipe. In the experiment corresponding to Figure 5 in the main text, at a gas flux of $70.1 \text{ mL}\cdot\text{min}^{-1}$, the total volume V_g of bubbles entering the pipe within 300 s is 350.5 mL. The electrical energy E_e output by the bubble energy harvesting device with a porous valve and the valveless device is 4.01 mJ and 8.54×10^{-1} mJ, respectively. The seawater density $\rho = 1.025\times 10^3 \text{ kg}\cdot\text{m}^{-3}$, acceleration of gravity $g = 9.8 \text{ m}\cdot\text{s}^{-2}$, and the height of the bubble rising pipe $L = 0.8 \text{ m}$. According to Equation (S15), the energy harvesting efficiencies of the bubble energy harvesting device equipped with the porous valve and the valveless device are calculated to be 0.14% and 0.03%, respectively.

The last paragraph of the section "Bubble Energy Harvesting Performance" in the main text has been revised as follows.

Table 1 provides a comparison between the bubble energy harvesting device based on the porous valve and existing devices in terms of application scenarios, energy conversion principles, and output energy density (electrical energy obtained from a unit volume of bubbles). Among them, the diving-surfacing device driven by chemical reaction relies on the buoyancy of carbon dioxide bubbles generated by calcite dissolution in acidic solutions to drive a wire to move in a magnetic field, thus being suitable for acidic solution environments. Its energy conversion efficiency from chemical energy to electrical energy is $1.5\times 10^{-5}\%$. The bubble energy harvesters based on electrostatic induction currently applies only to freshwater environments due to the problem of charge decay in salt solutions. The pipe fluid energy harvesting device driven by bubble buoyancy used in this work has no charge decay problem and is suitable for operation in seawater environments. According to the experimental results shown in Figure 5f, at a gas flux of $70.1 \text{ mL}\cdot\text{min}^{-1}$, the bubble energy harvesting device with no valve generates 8.54×10^{-1} mJ of electrical energy by utilizing 350.5 mL of bubbles over 300 s. The corresponding energy density and energy harvesting efficiency (Section 10 in the Supplementary Information) are $2.44 \text{ mJ}\cdot\text{L}^{-1}$ and 0.03%, respectively. Under the same conditions, the bubble energy harvesting device equipped with the porous valve outputs 4.01 mJ of electrical energy. The corresponding energy density and energy harvesting efficiency are $11.44 \text{ mJ}\cdot\text{L}^{-1}$ and 0.14%, respectively, both of which are 4.7 times those of the bubble energy harvesting device with no valve. The output energy density of the bubble energy harvesting device based on the porous valve has reached 6.7 times that of the existing bubble energy generator, and the energy harvesting efficiency of the porous valve-based device is four orders of magnitude higher than that of the diving-surfacing device driven by chemical reaction. These results of comparison prove that the bubble energy harvesting device based on the porous valve has favorable energy harvesting performance and high working efficiency.

The newly added relevant content in the Discussion section of the main text is as follows.

The proposed mechanical self-adaptive porous valve has a stable opening threshold inversely proportional to the size of its pore diameter. In the bubble buoyancy-driven pipe fluid energy harvesting systems, bubble rising pipe with different diameters and different types of turbine generators may require matching gas-liquid two-phase flows with different volume fractions of gas phase to achieve efficient actuation. Thus, prior to practical application, threshold matching tests should be conducted in advance. Rationally designing the opening threshold of the porous valve to adjust the gas-release volume per cycle contributes to further improving the rotational speed-flux coefficient and maximizing the bubble energy harvesting efficiency. Optimizing the gas-release volume per cycle by rationally designing the opening threshold of the porous valve contributes to further improving the rotational speed-flux coefficient and maximizing the bubble energy harvesting efficiency. Indeed, the mechanical self-adaptive porous valve serves as a universal passive bubble control device. Beyond the pipe fluid energy harvesting system used in this study, electrostatic and triboelectric bubble energy harvesters can similarly enhance energy harvesting efficiency by replacing bubble flow with slug flow to achieve rapid movement of bubbles along the surface of electret or triboelectric materials. Therefore, the proposed mechanical self-adaptive porous valve is also expected to be applied to other types of bubble energy harvesting systems, such as electrostatic and triboelectric systems, enhancing bubble energy harvesting performance by bubble accumulation and high-speed release.

7. The Demo in Fig. 6 did not have any scale bar. Based on the size of the device, I guess the bubbles generated by photosynthesis are quite tiny. The authors should discuss the effect of bubble size on the system.

Reply: Thank you for your valuable suggestion. Accordance to your suggestion, we have added a scale bar to Figure 6a. During the experiment, *Riccia fluitans* produces oxygen-rich bubbles through photosynthesis. Upon initial formation, bubbles are minuscule in volume, with buoyancy too weak to overcome adhesion forces, thus adhering to the surface of *Riccia fluitans*. As photosynthesis proceeds, the attached bubbles gradually expand. When their diameter reaches the millimeter scale, buoyancy overcomes adhesion, prompting detachment of bubbles from the plant surface. In the experiment, the released photosynthetic bubbles have a diameter of approximately 1–3 mm, which is similar to the size range of bubbles released by the needle of the bubble releaser.

Admittedly, the size of photosynthetic bubbles released by different species of aquatic plants may vary. Under external disturbances, the degree of dispersive movement exhibited by bubbles of different sizes also differs, which can affect the efficiency of bubble collection. We are currently developing a device for the efficient capture and convergence of multi-directional dispersed microbubbles in water. If you are interested in our research, please stay tuned for our subsequent research publications.

We have supplemented the description of experimental phenomena in the paragraph corresponding to Figure 6 in the main text as follows.

Figure 6a shows the accumulation process of ultralow-flux oxygen bubbles released by the underwater *Riccia fluitans* through photosynthesis into the gas reservoir of the porous valve. Upon initial formation, bubbles are minuscule in volume, with buoyancy too weak to overcome adhesion forces, thus adhering to the surface of *Riccia fluitans*. As photosynthesis proceeds, the attached bubbles gradually expand. When their diameter reaches the millimeter scale, buoyancy overcomes adhesion, prompting detachment of bubbles from the plant surface. In the experiment, the released photosynthetic bubbles have a diameter of approximately 1–3 mm.

In addition, we have supplemented the "Discussion" section with a discussion on the potential impact of bubble size on the system and improvement measures, as follows.

In the demonstration experiment of photosynthetic bubble energy harvesting, the gas reservoir of the porous valve collects bubbles released by aquatic plant photosynthesis through a funnel structure, achieving accumulation of bubble buoyancy potential energy. For future practical applications, considering various disturbance factors in submarine environment, such as ocean currents, tides fluctuations, marine organism movement, and subsea geological activities, microbubbles are prone to random dispersive motions. Photosynthetic bubbles released by different aquatic plants may vary in size. Under external disturbances, bubbles of different sizes exhibit differing degrees of random dispersive motions. These factors affect gas collection efficiency. To further enhance the environmental adaptability of the bubble energy harvesting device, a high-efficiency capture device for multidirectional dispersed microbubbles can be developed and installed beneath the porous valve. It contributes to improve the capture efficiency of microbubbles and the gas intake of energy harvesting devices, thereby boosting the power generation capacity at the source. In addition, considering potential fouling in marine environments, biomimetic principles can be adopted to design self-cleaning surface topography and microstructures based on the passive porous interface mechanical structure in the porous valve, further enhancing the durability of the porous valve in marine environments.

8. Life-span performance should be considered. Making allowance for the long-term use inside the seawater, the erosion of bubbles and water flow on devices, especially on the hydrophilic coatings, cannot be ignored.

Reply: Thank you for your valuable suggestions. The service life of the mechanical self-adaptive porous valve was taken into account during its design. To ensure the corrosion resistance of the porous valve in seawater, a corrosion-resistant polymer material—photopolymer resin—was selected as its material. To further demonstrate the stability of the hydrophilic and aerophobic effect on the porous plate surface, we conducted a continuous aeration impingement test by continuously releasing highly dense bubble plumes below the porous valve. The testing result has been added to

Section 12.2 of the Supplementary Information, which are as follows.

12.2 Bubble plume impingement

To demonstrate the stability of the aerophobic characteristics of the passive porous interface mechanical structure, an underwater bubble plume aeration impingement test was conducted on the porous valve. The experimental setup is shown in Figure S12(a), where a bubble plume releaser with a diameter of 9.6 cm is placed below the gas reservoir of the porous valve. The bubble plume generator is connected to an air pump via a silicone hose to release microbubbles with diameters on the micrometer scale into the water. The bubble plume releaser outputs highly dense microbubbles with a high specific surface area at a gas flux of $22.5 \text{ mL} \cdot \text{min}^{-1}$ to the gas reservoir of the porous valve. After the bubble plume impingement test was completed, the contact angle of a bubble on the surface of the porous plate in the porous valve was measured and presented in Figure S12(b). The open-circuit voltage of the energy harvesting device during the continuous impingement process of the bubble plume was measured and plotted in Figure S12(c).

Figure S12 Stability test of the bubble energy harvesting device based on the porous valve under bubble plume aeration impingement. (a) Aeration experimental setup; (b) Bubble contact angle on the surface of the passive porous interface mechanical structure in the porous valve after continuous scouring by bubble plumes; (c) Open-circuit voltage of the bubble energy harvesting device over 10000 s. The inset is the current waveforms in 1439–1467 s.

As shown in Figure S12(b), after 10000 s of continuous impingement by highly dense bubble plumes, the bubble contact angle on the surface of the passive porous interface mechanical structure

in seawater is 149° , which is close to the pre-impingement contact angle of 146° (Figure S2), demonstrating the stable aerophobic characteristics of the porous structure. As depicted in Figure S12(c), the open-circuit voltage of the bubble energy harvesting device based on the porous valve consistently exhibits stable periodic variations, corresponding to the cycle of bubble accumulation and high-speed release in the porous valve. During the 10000-second experiment, the porous valve released bubbles approximately every 17 seconds. The peak values of the open-circuit voltage over nearly 600 bubble-release cycles remain stable at around 0.2 V. Notably, the impingement intensity of bubble plumes in the actual ocean is far lower than the test conditions in this experiment. The gas flux of bubbles released per unit area of the seafloor is extremely low in real marine environments. For example, the gas fluxes at two observed bubble seepage points in the cold-spring activity area of Site F in the South China sea are 6.75 and 2.18 mL \cdot min $^{-1}$, with corresponding bubble diameters of 3.25 and 2.54 mm, respectively. This indicates that the gas flux, specific surface area, and impingement intensity of bubble plumes in actual marine environments are all much lower than the conditions of this experiment. Despite such harsh test conditions, the aerophobic characteristics of the passive porous interface mechanical structure and the electrical output of the energy harvesting device remain stable. These results validate that the proposed porous valve maintains a stable operational state, thereby ensuring that the bubble energy harvesting system based on the porous valve exhibits stable output performance.

Reviewer #3:

The manuscript titled "Passive porous interface-mechanical metamaterial relying on surface tension for efficient energy harvesting from ultralow-flux bubbles" presents a passive method that utilizes surface-tension-driven gas-liquid interface deformation to harvest energy from ultralow-flux underwater bubbles. Unlike conventional active systems, the design operates without external energy input, enabling self-adaptive bubble accumulation and release via Laplace pressure modulation. A porous valve based on this concept improves gas intake and energy conversion efficiency, with experimental results demonstrating effective energy harvesting from biologically generated bubbles. The study combines theoretical analysis and experimental validation, highlighting the potential of this approach for electrical energy production in support of subsea self-powered sensing and in situ autonomous exploration. Overall, the concept is interesting and can lead to a practical device.

Here are a few comments and questions:

1. In the abstract, "The maximum output power and electrical energy production are enhanced by factors of 36.63 and 16.42, respectively". Please clarify what baseline this enhancement is compared to in the abstract.

Reply: Thank you for your valuable suggestion. The multiples by which the bubble energy harvesting device with a porous valve improves in maximum output power and electrical energy

were obtained by comparing it with the device with no valve under the same experimental conditions. At a gas flux of $70.1 \text{ mL} \cdot \text{min}^{-1}$, the maximum output power of the porous valve-based device is 0.33 mW, which is 36.6 times that of the device with no valve ($9.01 \text{ } \mu\text{W}$). At a gas flux of $51.9 \text{ mL} \cdot \text{min}^{-1}$, the device with the porous valve generates 1.97 mJ of electrical energy over 300 s, which is 16.4 times the value (0.12 mJ) of the device with no valve. We have supplemented the baseline for comparison in the abstract.

The revised abstract is as follows.

Harvesting energy from subsea bubbles, such as those produced by photosynthesis of benthic plants or submarine methane seepage, is a promising solution for powering subsea environment perception devices, but the ultralow gas flux brings significant challenges. Herein, we propose a passive porous interface mechanical structure with self-adaptive mechanical properties and high gas permeability. Based on this structure, we develop a mechanical self-adaptive porous valve. It improves energy harvesting performance from ultralow-flux bubbles by controlling bubble accumulation and high-speed release. Unlike traditional active mechanical metamaterials, this passive design utilizes gas-liquid interface deformation (rather than metamaterial actuation) to generate self-adaptive Laplace pressure counteracting bubble buoyancy, and thus requires no external energy. The porous valve has a stable opening threshold inversely proportional to the size of its structural pore diameter. Compared with a bubble energy harvesting device with no valve, the instantaneous gas-intake rate of the device equipped with the porous valve is increased by one to four orders of magnitude, and the maximum output power and electrical energy production are enhanced by factors of 36.6 and 16.4, respectively. The energy of underwater biological metabolic gas with an ultralow flux ($28 \text{ } \mu\text{L} \cdot \text{min}^{-1}$) is effectively harvested and supplied to an underwater sensor. This work is expected to pave the way for subsea self-powered sensing and in situ autonomous exploration.

2. On page 6, the porous interface-mechanical metamaterial is treated with a hydrophilic modification. Could the authors comment on the long-term durability of the air-dried nano-silica coating under seawater conditions? In addition, beyond the porous material itself, was any surface treatment applied to the internal surface of the reservoir structure? As observed in Figure 6, numerous bubbles appear to be pinned on the inner wall of the reservoir inlet (the flared, funnel-shaped structure), which may reduce the efficiency of bubble collection.

Reply: Thank you for your question. The purpose of hydrophilic treatment on the passive porous interface mechanical structure is to make it exhibit aerophobic properties in water. To demonstrate the stability of the aerophobic characteristics of the porous structure, we have supplemented an experiment for stability test in Section 12.2 of the Supplementary Information.

In the experiment shown in Figure 6, we did not treat the funnel-shaped structure at the inlet of the gas reservoir. Indeed, during the experiment, some bubbles temporarily remained on the inner wall surface of the funnel before entering the gas reservoir. When the bubbles attached to the inner

wall of the funnel gradually merged into large bubbles, the tangential component of the bubble buoyancy force gradually exceeded the sliding resistance of the bubbles, eventually driving the bubbles on the inner wall surface to slide upward into the gas reservoir. Although this process does not affect the total volume of gas finally collected by the gas reservoir, it reduces the speed of gas collection. In future practical applications, the cone half-angle of the funnel can be appropriately reduced to enable bubbles attached to the inner wall of the funnel to enter the gas reservoir more quickly. In addition to using the funnel structure for gas collection, we are also developing a device for the efficient capture and convergence of multi-directional dispersed microbubbles in water to resist various disturbances that may exist in the actual marine environment. If you are interested in our research, please stay tuned for our subsequent research publications.

The newly added experimental content in the Supplementary Informations is as follows.

12.2 Bubble plume impingement

To demonstrate the stability of the aerophobic characteristics of the passive porous interface mechanical structure, an underwater bubble plume aeration impingement test was conducted on the porous valve. The experimental setup is shown in Figure S12(a), where a bubble plume releaser with a diameter of 9.6 cm is placed below the gas reservoir of the porous valve. The bubble plume generator is connected to an air pump via a silicone hose to release microbubbles with diameters on the micrometer scale into the water. The bubble plume releaser outputs highly dense microbubbles with a high specific surface area at a gas flux of $22.5 \text{ mL} \cdot \text{min}^{-1}$ to the gas reservoir of the porous valve. After the bubble plume impingement test was completed, the contact angle of a bubble on the surface of the porous plate in the porous valve was measured and presented in Figure S12(b). The open-circuit voltage of the energy harvesting device during the continuous impingement process of the bubble plume was measured and plotted in Figure S12(c).

As shown in Figure S12(b), after 10000 s of continuous impingement by highly dense bubble plumes, the bubble contact angle on the surface of the passive porous interface mechanical structure in seawater is 149° , which is close to the pre-impingement contact angle of 146° (Figure S2), demonstrating the stable aerophobic characteristics of the porous structure. As depicted in Figure S12(c), the open-circuit voltage of the bubble energy harvesting device based on the porous valve consistently exhibits stable periodic variations, corresponding to the cycle of bubble accumulation and high-speed release in the porous valve. During the 10000-second experiment, the porous valve released bubbles approximately every 17 seconds. The peak values of the open-circuit voltage over nearly 600 bubble-release cycles remain stable at around 0.2 V. Notably, the impingement intensity of bubble plumes in the actual ocean is far lower than the test conditions in this experiment. The gas flux of bubbles released per unit area of the seafloor is extremely low in real marine environments. For example, the gas fluxes at two observed bubble seepage points in the cold-spring activity area of Site F in the South China sea are 6.75 and $2.18 \text{ mL} \cdot \text{min}^{-1}$, with corresponding bubble diameters of 3.25 and 2.54 mm, respectively. This indicates that the gas flux, specific surface area, and

impingement intensity of bubble plumes in actual marine environments are all much lower than the conditions of this experiment. Despite such harsh test conditions, the aerophobic characteristics of the passive porous interface mechanical structure and the electrical output of the energy harvesting device remain stable. These results validate that the proposed porous valve maintains a stable operational state, thereby ensuring that the bubble energy harvesting system based on the porous valve exhibits stable output performance.

Figure S12 Stability test of the bubble energy harvesting device based on the porous valve under bubble plume aeration impingement. (a) Aeration experimental setup; (b) Bubble contact angle on the surface of the passive porous interface mechanical structure in the porous valve after continuous scouring by bubble plumes; (c) Open-circuit voltage of the bubble energy harvesting device over 10000 s. The inset is the current waveforms in 1439–1467 s.

We have added the relevant content in the "Discussion" section of the main text as follows.

In the demonstration experiment of photosynthetic bubble energy harvesting, the gas reservoir of the porous valve collects bubbles released by aquatic plant photosynthesis through a funnel structure, achieving accumulation of bubble buoyancy potential energy. For future practical applications, considering various disturbance factors in submarine environment, such as ocean currents, tides fluctuations, marine organism movement, and subsea geological activities, microbubbles are prone to random dispersive motions. Photosynthetic bubbles released by different aquatic plants may vary in size. Under external disturbances, bubbles of different sizes exhibit differing degrees of random dispersive motions. These factors affect gas collection efficiency. To

further enhance the environmental adaptability of the bubble energy harvesting device, a high-efficiency capture device for multidirectional dispersed microbubbles can be developed and installed beneath the porous valve. It contributes to improve the capture efficiency of microbubbles and the gas intake of energy harvesting devices, thereby boosting the power generation capacity at the source. In addition, considering potential fouling in marine environments, biomimetic principles can be adopted to design self-cleaning surface topography and microstructures based on the passive porous interface mechanical structure in the porous valve, further enhancing the durability of the porous valve in marine environments.

3. Based on the introduction (page 1), the observed gas fluxes in actual marine environments are on the order of 6.75 mL/min and 2.18 mL/min, which are lower than the experimental condition of 70.1 mL/min (page 9). Could the authors provide an estimate of the energy that could be generated under real-world conditions using this passive porous interface-mechanical metamaterial over a period of operation?

Reply: Thank you for your question and suggestion. According to your suggestion, we estimated the annual electricity generation potential based on the output energy density of the bubble energy harvesting device with a porous valve and the gas flux data of a bubble seepage point in the cold-spring activity area of Site F in the South China sea from Reference 30. The corresponding calculation process and result were added to Section 11 of the Supplementary Information, which are as follows.

11. Estimate of electricity generation potential

For a bubble with volume V_0 released from a seabed at a depth h_0 below the sea surface, according to the ideal gas equation, when the temperature and the amount of substance are constant, the gas pressure of the bubble is inversely proportional to its volume. During the ascent of the bubble in water, the gas pressure equilibrates with the surrounding seawater pressure. The decrease in the surrounding seawater pressure causes the gas pressure within the bubble to gradually decrease, leading to the expansion of the bubble volume. When the bubble rises to a water depth h , its volume $V(h)$ can be expressed as

$$V(h) = \frac{(\rho g h_0 + p_{\text{atm}}) V_0}{\rho g h + p_{\text{atm}}}, \quad (\text{S16})$$

where $\rho = 1.025 \times 10^3 \text{ kg} \cdot \text{m}^{-3}$ is the density of seawater around the bubble, $g = 9.8 \text{ m} \cdot \text{s}^{-2}$ is the acceleration of gravity, and $p_{\text{atm}} = 1.01 \times 10^5 \text{ Pa}$ is the atmospheric pressure. For the bubble energy harvesting device based on the porous valve, bubbles rise in the pipe in the form of slug flow shown in Figure 5b of the main text, each being a long baculiform bubble sufficient to fill the pipe diameter. The bubbles only contact water at the front and rear ends, leading to low mass and heat transfer efficiencies. Therefore, dissolution loss and temperature change of bubbles during ascent in the pipe

are negligible. According to the results in Table 1 of the main text, the output energy density (the electrical energy obtained from a unit volume of bubbles) of the bubble energy harvesting device equipped with the porous valve is $11.44 \text{ mJ}\cdot\text{L}^{-1}$ within a bubble rising height of 0.8 m. Accordingly, the expected electrical energy generated by a bubble with volume V_0 rising from an underwater depth h_0 to the sea surface is

$$E_e = \int_{h_0}^0 \frac{E_d}{0.8} V(h) d(h_0 - h) \quad (\text{S17})$$

Substituting Equation (S16) into Equation S17 gives

$$E_e = \frac{1.25E_d(\rho gh_0 + p_{\text{atm}})V_0}{\rho g} \ln\left(1 + \frac{\rho gh_0}{p_{\text{atm}}}\right) \quad (\text{S18})$$

Taking a bubble seepage point in the cold-spring activity area of Site F in the South China sea as an example to estimate the power generation potential. The bubble seepage point is located on the 1120 m deep seafloor, with a gas flux of $6.75 \text{ mL}\cdot\text{min}^{-1}$, and the annual bubble release volume can be calculated to be approximately $3.55 \times 10^3 \text{ L}$. By substituting the relevant data ($h_0 = 1120 \text{ m}$ and $V_0 = 3.55 \times 10^3 \text{ L}$) into Equation (S18), the annual power generation is calculated to be $2.71 \times 10^5 \text{ J}$.

4. What is the effective range of pore sizes that allows the proposed concept to function reliably? As the pore size decreases, is there a risk that the accumulated gas pressure could displace the liquid back into the bulk rather than allowing the gas to pass through the porous interface?

Reply: Thank you for your question. According to the experimental results, the effective range of pore diameter for the mechanical self-adaptive porous valve to function reliably is 0.8–1.7 mm. We attempted to expand the range of pore diameter and observed the following experimental phenomena: (1) When the pore diameter of the porous valve exceeds 1.7 mm, the volume of gas that the porous valve can accumulate is too small to read the opening threshold, and the bubble-accumulation capacity almost disappears. (2) When the pore diameter is below 0.8 mm, processing errors significantly reduce the uniformity of pore diameters among different micropores on the porous plate. The accumulated gas can only be slowly released from the individual micropore with the largest actual pore diameter on the porous plate surface, leading to a significant decrease in the gas-release rate and making it difficult to achieve high-speed release of the accumulated gas.

According to interface mechanical theory, the gas pressure on one side of a horizontal gas-liquid interface (with a theoretically infinite curvature radius) is in equilibrium with the liquid pressure on the other side, resulting in no Laplace pressure difference. The pressure p_g of the accumulated gas varies with the liquid pressure p_d below the gas, maintaining equilibrium ($p_g = p_d$), as mentioned in the text above Equation (5) in the main text. Therefore, the pressure p_g of the accumulated gas cannot exceed the liquid pressure p_d , eliminating the risk that the accumulated gas pressure could push the liquid back into the bulk in physical principle.

The seventh paragraph of the section "Threshold characteristic" in the main text has been

revised as follows.

The decreasing tendency of the opening threshold as the pore diameter increases means that the gas-storage capacity of the porous valve decreases with the pore diameter. As shown in Figure 4c, the porous valve with a larger pore diameter has a weaker gas-storage capacity (lower maximum gas-storage volume), and thus the volume of gas released after each open event is smaller. Similar results are found in Figure 4d, where the duration of each bubble-release stage decreases as the pore diameter increases. According to the gas-release volume and duration, the gas-release rate of the bubble-release stage in each cycle is calculated and listed in Figure 4e. Within the pore diameter range of 0.8–1.7 mm, the mean value of the gas-release rate of the porous valve during the bubble-release stage shows a gradually increasing trend as the pore diameter increases. When the pore diameter exceeds 1.7 mm, as shown in Figure S8 of the Supplementary Information, the volume of gas that the porous valve can accumulate is too small to reach the opening threshold, and the bubble-accumulation capacity almost disappears. When the pore diameter is below 0.8 mm, processing errors significantly reduce the uniformity of pore diameters among different micropores on the porous plate. The accumulated gas can only be slowly released from the individual micropore with the largest actual pore diameter on the porous plate surface, leading to a significant decrease in the gas-release rate and making it difficult to achieve high-speed release of the accumulated gas. In practical applications, if it is necessary to further enhance the gas-storage capacity of the porous valve without reducing the pore diameter, it can be achieved by increasing the cross-sectional area of the gas reservoir.

5. On page 9, the authors state that Figure 4e indicates an increasing trend in gas-release rate with increasing pore diameter. However, the error bars shown in Figure 4e suggest that the differences in gas-release rate between pore sizes in the range of 1.3 to 1.7 mm may not be significant, making the overall trend less clear. Could the authors clarify whether the observed trend can be considered general across all experimental conditions?

Reply: Thank you for your question. To further validate the generality of the trend that gas-release rate increases with pore diameter, we repeated experiments on the porous valves with different pore diameters under an ultralow-flux condition of $11.0 \text{ mL} \cdot \text{min}^{-1}$. Using a lower gas flux can significantly reduce the disturbance of the gas inlet behavior below the gas reservoir on the process of bubble release, thereby revealing more general laws. The experimental results have been supplemented into Section 8.1 of the Supplementary Information, which are as follows.

In the main text, Figure 4e shows the variation trend of the gas-release rate of the porous valves with pore diameter at an environmental gas flux of $70.1 \text{ mL} \cdot \text{min}^{-1}$. To minimize the influence of the gas-intake behavior on the process of bubble release, eight repetitive experiments were conducted on the porous valves with different pore diameters under the minimum gas flux ($11.0 \text{ mL} \cdot \text{min}^{-1}$)

provided by the experimental setup in Figure S4. The corresponding gas-release rates are presented in Table S5. The experimental results show that the average gas-release rate of porous valves during the bubble-release stage still exhibits an increasing trend with the increase in pore diameter.

Table S5 Gas-release rate during the bubble-release stage of the porous valves with different pore diameters in each cycle at a low environmental gas flux of $11.0 \text{ mL} \cdot \text{min}^{-1}$ (data unit: $\text{L} \cdot \text{min}^{-1}$)

Pore diameters d_u (mm)	1st	2nd	3rd	4th	5th	6th	7th	8th	Mean value	Standard deviation
0.8	1.12	1.13	1.20	1.15	1.05	1.25	1.05	1.14	1.14	0.07
1.0	1.19	1.21	1.35	1.13	1.15	1.39	1.47	1.56	1.30	0.15
1.3	1.56	1.43	1.46	1.46	1.36	1.49	1.49	1.58	1.48	0.07
1.5	1.95	2.03	1.66	1.90	1.87	1.95	1.87	1.98	1.90	0.11
1.7	2.26	2.14	2.26	2.06	1.94	2.03	2.19	1.96	2.10	0.12

6. On page 12, Table 1 lists the gas flux used in this work as $70.1 \text{ mL}/\text{min}$. If applicable, could the authors include the gas flux values used in other referenced works in the table for better comparison?

Reply: Thank you for your valuable suggestion. According to your suggestion, we have supplemented Table 1 with a more comprehensive comparison between the gas-liquid two-phase flow energy harvesting device based on the proposed porous valve and various existing bubble energy harvesters, including gas flux values.

The revised Table 1 and corresponding text description are as follows.

Table 1 provides a comparison between the bubble energy harvesting device based on the porous valve and existing devices in terms of application scenarios, energy conversion principles, and output energy density (electrical energy obtained from a unit volume of bubbles). Among them, the diving-surfacing device driven by chemical reaction relies on the buoyancy of carbon dioxide bubbles generated by calcite dissolution in acidic solutions to drive a wire to move in a magnetic field, thus being suitable for acidic solution environments. Its energy conversion efficiency from chemical energy to electrical energy is $1.5 \times 10^{-5}\%$. The bubble energy harvesters based on electrostatic induction currently applies only to freshwater environments due to the problem of charge decay in salt solutions. The pipe fluid energy harvesting device driven by bubble buoyancy used in this work has no charge decay problem and is suitable for operation in seawater environments. According to the experimental results shown in Figure 5f, at a gas flux of $70.1 \text{ mL} \cdot \text{min}^{-1}$, the bubble energy harvesting device with no valve generates $8.54 \times 10^{-1} \text{ mJ}$ of electrical energy by utilizing 350.5 mL of bubbles over 300 s . The corresponding energy density and energy harvesting efficiency (Section 10 in the Supplementary Information) are $2.44 \text{ mJ} \cdot \text{L}^{-1}$ and 0.03% , respectively. Under the same conditions, the bubble energy harvesting device equipped with the porous valve outputs 4.01 mJ of electrical energy. The corresponding energy density and energy harvesting efficiency are $11.44 \text{ mJ} \cdot \text{L}^{-1}$ and 0.14% , respectively, both of which are 4.7 times those of

the bubble energy harvesting device with no valve. The output energy density of the bubble energy harvesting device based on the porous valve has reached 6.7 times that of the existing bubble energy generator, and the energy harvesting efficiency of the porous valve-based device is four orders of magnitude higher than that of the diving-surfacing device driven by chemical reaction. These results of comparison prove that the bubble energy harvesting device based on the porous valve has favorable energy harvesting performance and high working efficiency.

Table 1 Comparison between existing researches and the bubble energy harvesting device (BEH) with the proposed mechanical self-adaptive porous valve (MSPV)

Applicable scene	References	Energy conversion principle	Volume of bubbles V_b (mL)	Gas flux Q_g (mL·min ⁻¹)	Electrical energy output E (mJ)	Energy density E_d (mJ·L ⁻¹)
Acidic solution	Diving-surfacing device driven by chemical reaction	Chemical reactions and electromagnetic induction	176	35.2	1.8×10^{-5}	1.02×10^{-1}
Fresh water	Fluidic electricity generator	Electrostatic induction	1.14	30	9.38×10^{-7}	8.23×10^{-4}
	Dual-electrode electrets	Electrostatic induction	0.189	10	7.0×10^{-5}	3.70×10^{-1}
	Multi-electrode electrets	Electrostatic induction	0.25	10	3.9×10^{-5}	1.56×10^{-1}
	Bubble energy generator	Electrostatic induction	0.1	10	1.72×10^{-4}	1.72
	Lubricant-impregnated bubble energy generator	Electrostatic induction	0.1	10	1.06×10^{-4}	1.06
Seawater	Broadband rotary hybrid generator	Electromagnetic induction and piezoelectric effect	1500	/	2.15	1.43
	Biomimetic cactus bubble energy harvester	Electromagnetic induction	18.7	16.4	1.17×10^{-1}	6.26
	BEH without the MSPV	Electromagnetic induction	350.5	70.1	8.54×10^{-1}	2.44
	BEH with the MSPV	Electromagnetic induction	350.5	70.1	4.01	11.44

Reviewer #4:

Reply: Thank you for reviewing our manuscript.